# Long Live the Lottery: The Existence of Winning Tickets in Lifelong Learning

**Tianlong Chen[1*], Zhenyu Zhang[2*], Sijia Liu[3,4], Shiyu Chang[4], Zhangyang Wang[1]**
[1]University of Texas at Austin,[2]University of Science and Technology of China
[3]Michigan State University, [4]MIT-IBM Watson AI Lab, IBM Research
`{tianlong.chen,atlaswang}@utexas.edu, zzy19969@mail.ustc.edu.cn`
`liusiji5@msu.edu, shiyu.chang@ibm.com`

## Abstract

The lottery ticket hypothesis states that a highly sparsified sub-network can be trained in isolation, given the appropriate weight initialization. This paper extends that hypothesis from one-shot task learning, and demonstrates for the first time that such extremely compact and independently trainable sub-networks can be also identified in the lifelong learning scenario, which we call *lifelong tickets*. We show that the resulting lifelong ticket can further be leveraged to improve the performance of learning over continual tasks. However, it is highly non-trivial to conduct network pruning in the lifelong setting. Two critical roadblocks arise: i) As many tasks now arrive sequentially, finding tickets in a greedy weight pruning fashion will inevitably suffer from the intrinsic bias, that the earlier emerging tasks impact more; ii) As lifelong learning is consistently challenged by catastrophic forgetting, the compact network capacity of tickets might amplify the risk of forgetting. In view of those, we introduce two pruning options, e.g., *top-down* and *bottom-up*, for finding lifelong tickets. Compared to the top-down pruning that extends vanilla (iterative) pruning over sequential tasks, we show that the bottom-up one, which can dynamically shrink and (re-)expand model capacity, effectively avoids the undesirable excessive pruning in the early stage. We additionally introduce *lottery teaching* that further overcomes forgetting via knowledge distillation aided by external unlabeled data. Unifying those ingredients, we demonstrate the existence of very competitive lifelong tickets, e.g., achieving $3 - 8\%$ of the dense model size with even higher accuracy, compared to strong class-incremental learning baselines on CIFAR-10/CIFAR-100/Tiny-ImageNet datasets. Codes available at `https://github.com/VITA-Group/Lifelong-Learning-LTH`.

## 1 Introduction

The lottery ticket hypothesis (LTH) (Frankle & Carbin, 2019) suggests the existence of an extremely sparse sub-network, within an overparameterized dense neural network, that can reach similar performance as the dense network when trained in isolation with proper initialization. Such a sub-network together with the used initialization is called a *winning ticket* (Frankle & Carbin, 2019). The original LTH studies the sparse pattern of neural networks with a single task (classification), leaving the question of generalization across multiple tasks open. Following that, a few works (Morcos et al., 2019; Mehta, 2019) have explored LTH in transfer learning. They study the transferability of a winning ticket found in a source task to another target task. This provides insights on *one-shot transferability* of LTH. In parallel, lifelong learning not only suffers from notorious catastrophic forgetting over sequentially arriving tasks but also often comes at the price of increasing model capacity. With those in mind, we ask a much more ambitious question:

*Does LTH hold in the setting of lifelong learning when different tasks arrive sequentially?*

Intuitively, a desirable "ticket" sub-network in lifelong learning (McCloskey & Cohen, 1989; Parisi et al., 2019) needs to be: 1) independently trainable, same as the original LTH; 2) trained to perform

---
[*]Equal Contribution.

competitively to the dense lifelong model, including both maintaining the performance of previous tasks, and quickly achieving good generalization at newly added tasks; 3) found online, as the tasks sequentially arrive without any pre-assumed order. We define such a sub-network with its initialization as a *lifelong lottery ticket*.

This paper seeks to locate the lifelong ticket in *class-incremental learning (CIL)* (Wang et al., 2017; Rosenfeld & Tsotsos, 2018; Kemker & Kanan, 2017; Li & Hoiem, 2017; Belouadah & Popescu, 2019; 2020), a popular, realistic and challenging setting of lifelong learning. A natural idea to extend the original LTH is to introduce sequential pruning: we continually prune the dense network until the desired sparsity level, as new tasks are incrementally added. However, we show that the *direct* application of the iterative magnitude pruning (IMP) used in LTH *fails* in the scenario of CIL since the pruning schedule becomes critical when tasks arrive sequentially. To circumvent this challenge, we generalize IMP to incorporate a curriculum pruning schedule. We term this technique *top-down lifelong pruning*. When the total number of tasks is pre-known and small, then with some "lottery" initialization (achieved by rewinding (Frankle et al., 2019) or similar), we find that the pruned sparse ticket can be re-trained to similar performance as the dense network. However, if the number of tasks keeps increasing, the above ticket will soon witness performance collapse as its limited capacity cannot afford the over-pruning.

The limitation of top-down lifelong pruning reminds us of *two* unique dilemmas that might challenge the validity of lifelong tickets. *i) Greedy weight pruning v.s. all tasks' performance*: While the sequential pruning has to be performed online, its greedy nature inevitably biases against later arriving tasks, as earlier tasks apparently will contribute to shaping the ticket more (and might even use up the sparsity budget). *ii) Catastrophic forgetting v.s. small ticket size:* To overcome the notorious catastrophic forgetting (McCloskey & Cohen, 1989; Tishby & Zaslavsky, 2015), many lifelong learning models have to frequently consolidate weights to carefully re-assign the model capacity (Zhang et al., 2020) or even grow model size as tasks come in (Wang et al., 2017). Those seem to contradict our goal of pruning by seeing more tasks.

To address the above two limitations, we propose a novel *bottom-up lifelong pruning* approach, which allows for re-growing the model capacity to compensate for any excessive pruning. It therefore flexibly calibrates between increasing and decreasing tickets throughout the entire learning process, alleviating the intrinsic greedy bias caused by the top-down pruning. We additionally introduce *lottery teaching* to overcome forgetting, which regularizes previous task models' soft logit outputs by using free unlabeled data. That is inspired by lifelong knowledge preservation techniques (Castro et al., 2018; He et al., 2018; Javed & Shafait, 2018; Rebuffi et al., 2017).

For validating our proposal, we conduct extensive experiments on CIFAR-10, CIFAR-100, and Tiny-ImageNet datasets for class-incremental learning (Rebuffi et al., 2017). The results demonstrate the existence and the high competitiveness of *lifelong tickets*. Our best lifelong tickets (found by bottom-up pruning and lottery teaching) achieve comparable or better performance across all sequential tasks, with as few as $3.64\%$ parameters, compared to state-of-the-art dense models. Our contributions can be summarized as:

- The problem of lottery tickets is formulated and studied in lifelong learning (class incremental learning) for the first time.
- Top-down pruning: a generalization of iterative weight magnitude pruning used in the original LTH over continual learning tasks.
- Bottom-up pruning: a novel pruning method, which is unique to allow for re-growing model capacity, throughout the lifelong process.
- Extensive experiments and analyses demonstrating the promise of lifelong tickets, in achieving superior yet extremely light-weight lifelong learners.

## 2 RELATED WORK

**Lifelong Learning** A lifelong learning system aims to continually learn sequential tasks and accommodate new information while maintaining previously learned knowledge (Thrun & Mitchell, 1995). One of its major challenges is called catastrophic forgetting (McCloskey & Cohen, 1989; Kirkpatrick et al., 2017; Hayes & Kanan, 2020), i.e., the network cannot maintain expertise on tasks that they have not experienced for a long time.

This paper's study subject is class-incremental learning (CIL) (Rebuffi et al., 2017; Elhoseiny et al., 2018): a popular, realistic, albeit challenging setting of lifelong learning. CIL requires the model to recognize new classes emerging over time while maintaining recognizing ability over old classes without access to the previous data. Typical solutions are based on regularization (Li & Hoiem, 2017; Kirkpatrick et al., 2017; Zenke et al., 2017; Aljundi et al., 2018a; Ebrahimi et al., 2019), for example, knowledge distillation (Hinton et al., 2015) is a common regularizer to inherit previous knowledge through preserving soft logits of those samples (Li & Hoiem, 2017) while learning new tasks. Besides, several approaches are learning with memorized data (Castro et al., 2018; Javed & Shafait, 2018; Rebuffi et al., 2017; Belouadah & Popescu, 2019; 2020; Lopez-Paz & Ranzato, 2017; Chaudhry et al., 2018). And some generative lifelong learning methods (Liu et al., 2020; Shin et al., 2017) mitigate catastrophic forgetting by generating simulated data of previous tasks. There also exist a few architecture-manipulation-based lifelong learning methods (Rajasegaran et al., 2019; Aljundi et al., 2018b; Hung et al., 2019; Abati et al., 2020; Rusu et al., 2016; Kemker & Kanan, 2017), while their target is dividing a dense model into task-specific parts for lifelong learning, rather than localizing sparse networks and the lottery tickets.

**Pruning and Lottery Ticket Hypothesis**  It is well-known that deep networks could be pruned of excess capacity (LeCun et al., 1990b). Pruning algorithms can be categorized into unstructured (Han et al., 2015b; LeCun et al., 1990a; Han et al., 2015a) and structured pruning (Liu et al., 2017; He et al., 2017; Zhou et al., 2016). The former sparsifies weight elements based on magnitudes, while the latter removes network sub-structures such as channels for more hardware friendliness.

LTH (Frankle & Carbin, 2019) advocates the existence of an independently trainable sparse sub-network from a dense network. In addition to image classification (Frankle & Carbin, 2019; Liu et al., 2019; Wang et al., 2020; Evci et al., 2019; Frankle et al., 2020; Savarese et al., 2020; You et al., 2020; Ma et al., 2021; Chen et al., 2020a), LTH has been explored widely in numerous contexts, such as natural language processing (Gale et al., 2019; Chen et al., 2020b), reinforcement learning (Yu et al., 2019), generative adversarial networks (Chen et al., 2021b), graph neural networks (Chen et al., 2021a), and adversarial robustness (Cosentino et al., 2019). Most of them adopt unstructured weight magnitude pruning (Han et al., 2015a; Frankle & Carbin, 2019) to obtain the ticket, which we also follow in this work. (Frankle et al., 2019) analyzes large models and datasets, and presents a rewinding technique that re-initializes ticket training from the early training stage rather than from scratch. (Renda et al., 2020) further compares different retraining techniques and endorses the effectiveness of rewinding. (Mehta, 2019; Morcos et al., 2019; Desai et al., 2019) pioneer to study the transferability of the ticket identified on one source task to another target task, which delivers insights on *one-shot transferability* of LTH.

One latest work (Golkar et al., 2019) aimed at lifelong learning in fixed-capacity models based on pruning neurons of low activity. The authors observed that a controlled way of "graceful forgetting" after training each task can regain network capacity for new tasks, meanwhile not suffering from forgetting. Sokar et al. (2020) further compresses the sparse connections of each task during training, which reduces the interference between tasks and alleviates forgetting.

## 3 LOTTERY TICKET FROM SINGLE-TASK LEARNING TO CIL

### 3.1 PROBLEM SETUP

In CIL, a model continuously learns from a sequential data stream in which *new* tasks (namely, classification tasks with new classes) are added over time, as shown in Figure 1. At the inference stage, the model can operate without having access to the information of task IDs. Following (Castro et al., 2018; He et al., 2018; Rebuffi et al., 2017), a handful of samples from previous classes are stored in a fixed memory buffer.

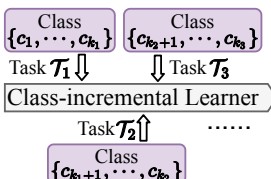

Figure 1: Basic CIL Setting.

More formally, let $\mathcal{T}_1, \mathcal{T}_2, \cdots$ represent a sequence of tasks, and the $i$th task $\mathcal{T}_i$ contains data that fall into $(k_i - k_{i-1})$ classes $\mathcal{C}_i = \{c_{k_{i-1}+1}, c_{k_{i-1}+2}, \cdots, c_{k_i}\}$, where $k_0 = 0$ by convention. Let $\Theta^{(i)} = \{\boldsymbol{\theta}^{(i)}, \boldsymbol{\theta}_c^{(i)}\}$ denote the model of the learner used at task $i$, where $\boldsymbol{\theta}^{(i)}$ corresponds to the *base* model cross all tasks from $\mathcal{T}_1$ to $\mathcal{T}_i$, and $\boldsymbol{\theta}_c^{(i)}$ denotes the *task-specific* classification head at $\mathcal{T}_i$.

Thus, the size of $\boldsymbol{\theta}^{(i)}$ is fixed, but the dimension of $\boldsymbol{\theta}_{\mathrm{c}}^{(i)}$ aligns with the number of classes, which have been seen at $\mathcal{T}_i$. In general, the learner has access to two types of information at task $i$: the current training data $\mathcal{D}^{(i)}$, and certain previous information $\mathcal{P}^{(i)}$. The latter includes a small amount of data from previous tasks $\{\mathcal{T}_j\}_1^{i-1}$ stored in the memory buffer, and the previous model $\Theta^{(i-1)}$ at task $\mathcal{T}_{i-1}$. This is commonly used to overcome the catastrophic forgetting issue of the current task $i$ against the previous tasks. Based on the aforementioned setting, we state the CIL problem as below.

**Problem of CIL.** *At the current task $i$, we aim to learn a full model $\Theta^{(i)} = \{\boldsymbol{\theta}^{(i)}, \boldsymbol{\theta}_{\mathrm{c}}^{(i)}\}$ based on the information $(\mathcal{D}^{(i)}, \mathcal{P}^{(i)})$ such that $\Theta^{(i)}$ not only (I) yields the generalization ability to the newly added data at task $\mathcal{T}_i$ but also (II) does not lose its power to the previous tasks $\{\mathcal{T}_j\}_1^{i-1}$.*

We note that the aforementioned problem statement applies to CIL with any fixed-length learning period. That is, for $n$ time stamps (one task per time), the validity of the entire trajectory $\{\Theta^{(i)}\}_1^n$ is justified by each $\Theta^{(i)}$ from the CIL criteria *(I)* and *(II)* stated in 'Problem of CIL'.

### 3.2 Lifelong lottery tickets

It was shown by LTH (Frankle & Carbin, 2019) that a standard (unstructured) pruning technique can uncover the so-called *winning ticket*, namely, a sparse sub-network together with proper initialization that can be trained in isolation and reach similar performance as the dense network. In this paper, we aim to prune the base model $\boldsymbol{\theta}^{(i)}$ over time. And we ask: *Do there exist winning tickets in lifelong learning? If yes, how to obtain them?* To answer these questions, a prerequisite is to define the notion of lottery tickets in lifelong learning, which we call *lifelong lottery tickets*.

Following LTH (Frankle & Carbin, 2019), a lottery ticket consists of two parts: 1) a binary mask $\mathbf{m} \in \{0,1\}^{\|\boldsymbol{\theta}^{(i)}\|_0}$ obtained from a one-shot or iterative pruning algorithm, and 2) initial weights or rewinding weights $\boldsymbol{\theta}_0$. The ticket $(\mathbf{m}, \boldsymbol{\theta}_0)$ is a winning ticket if training the subnetwork $\mathbf{m} \odot \boldsymbol{\theta}_0$ ($\odot$ denotes element-wise product), identified by the sparse pattern $\mathbf{m}$ with initialization $\boldsymbol{\theta}_0$, wins the initialization lottery to match the performance of the original (fully trained) network. In CIL, at the presence of sequential tasks $\{\mathcal{T}^{(i)}\}_{i=1,2,\ldots}$, we define *lifelong lottery tickets* $(\mathbf{m}^{(i)}, \boldsymbol{\theta}_0^{(i)})$ recursively from the perspective of dynamical system:

$$\mathbf{m}^{(i)} = \mathbf{m}^{(i-1)} + \mathcal{A}(\mathcal{D}^{(i)}, \mathcal{P}^{(i)}, \mathbf{m}^{(i-1)}), \quad \text{and} \quad \boldsymbol{\theta}_0^{(i)} \in \{\boldsymbol{\theta}^{(0)}, \boldsymbol{\theta}_{\mathrm{rw}}^{(i)}\}, \tag{1}$$

where $\mathcal{A}$ denotes a pruning algorithm used at the current task $\mathcal{T}^{(i)}$ based on the information $\mathcal{D}^{(i)}$, $\mathcal{P}^{(i)}$ and $\mathbf{m}^{(i-1)}$, $\boldsymbol{\theta}^{(0)}$ denotes the initialization prior to training the model at $\mathcal{T}^{(1)}$, and $\boldsymbol{\theta}_{\mathrm{rw}}^{(i)}$ denotes a rewinding point at $\mathcal{T}^{(i)}$. In Eq. (1), we interpret the (non-trivial) pruning operation $\mathcal{A}$ by weight perturbations, with values drawn from $\{-1, 0, 1\}$, to the previous binary mask. Here $-1$ denotes the removal of a weight, $0$ signifies to keep a weight intact, and $1$ represents the addition of a weight. Moreover, the introduction of weight rewinding is spurred by the so-called *rewinding ticket* (Renda et al., 2020; Frankle et al., 2020). For example, if $\boldsymbol{\theta}_{\mathrm{rw}}^{(i)} = \boldsymbol{\theta}^{(i-1)}$, then we pick the model weights learnt at the previous task $\mathcal{T}^{(i-1)}$ to initialize the training at $\mathcal{T}^{(i)}$. We also note that $\boldsymbol{\theta}^{(0)}$ can be regarded as the point rewound to the earliest stage of the lifelong learning. Based on Eq. (1), we then state the definition of *winning tickets* in CIL.

**Lifelong winning tickets.** *Given a sequence of tasks $\{\mathcal{T}_i\}_1^n$, the lifelong lottery tickets $\{(\mathbf{m}^{(i)}, \boldsymbol{\theta}_0^{(i)})\}_1^n$ given by (1) are winning tickets if they can be trained in isolation to match the CIL performance (i.e., criteria I and II) of the corresponding full model $\{\boldsymbol{\theta}^{(i)}\}_1^n$, where $n \in \mathbb{N}^+$.*

In the next section, we will design the lifelong pruning algorithm $\mathcal{A}$, together with ticket initialization schemes formulated in Eq. (1)

## 4 Proposed Pruning Method to Find Lifelong Winning Tickets

### 4.1 Revisiting IMP over sequential tasks: Top-down (TD) pruning

In order to find the potential tickets at the current task $\mathcal{T}^{(i)}$, it is natural to specify $\mathcal{A}$ in Eq. (1) as the iterative magnitude pruning (IMP) algorithm (Han et al., 2015a) to prune the model from

$\mathbf{m}^{(i-1)} \odot \boldsymbol{\theta}^{(i-1)}$. Following (Frankle & Carbin, 2019; Renda et al., 2020), IMP iteratively prunes $p^{\frac{1}{n^{(i)}}}$ (%) non-zero weights of $\mathbf{m}^{(i-1)} \odot \boldsymbol{\theta}^{(i-1)}$ over $n^{(i)}$ rounds at $\mathcal{T}^{(i)}$. Thus, the number of non-zero weights in the obtained mask $\mathbf{m}^{(i)}$ is given by $((1 - p^{\frac{1}{n^{(i)}}})^{n^{(i)}} \cdot \|\mathbf{m}^{(i-1)}\|_0)$. *However*, in the application of IMP to the sequential tasks $\{\mathcal{T}^{(i)}\}$, we find that the schedule of IMP over sequential tasks, in terms of $\{n^{(i)}\}$, is critical to make pruning successful in lifelong learning. We refer readers to Appendix A2.1 for detailed justifications.

**Curriculum schedule of TD pruning is a key to success**    The conventional method is to set $\{n^{(i)}\}$ as a uniform schedule, namely, IMP prunes a fixed portion of non-zeros at each task. However, this direct application fails quickly as the number of incremental tasks increases, implying that "not all tasks are created equal" in the learning/pruning schedule. Inspired by the recent observation that training with more classes helps consolidate a more robust sparse model (Morcos et al., 2019), we propose a curriculum pruning schedule, in which IMP is conducted more aggressively for new tasks arriving later, with $n^{(i)} \geq n^{(i-1)}$, until reaching the desired sparsity. For example, if there are 12 times of pruning on five sequentially arrived tasks, we arrange them in a linearly increasing way, i.e., $(\mathcal{T}_1:1, \mathcal{T}_2:1, \mathcal{T}_3:2, \mathcal{T}_4:3, \mathcal{T}_5:5)$. Note that TD pruning relies on the heuristic curriculum schedule, and thus inevitably greedy and suboptimal over continual learning tasks. In what follows, we propose a more advanced pruning scheme, bottom-up (BU) pruning, that obeys a different principle of design.

## 4.2    BOTTOM-UP (BU) LIFELONG PRUNING: AN ADVANCED SCHEME

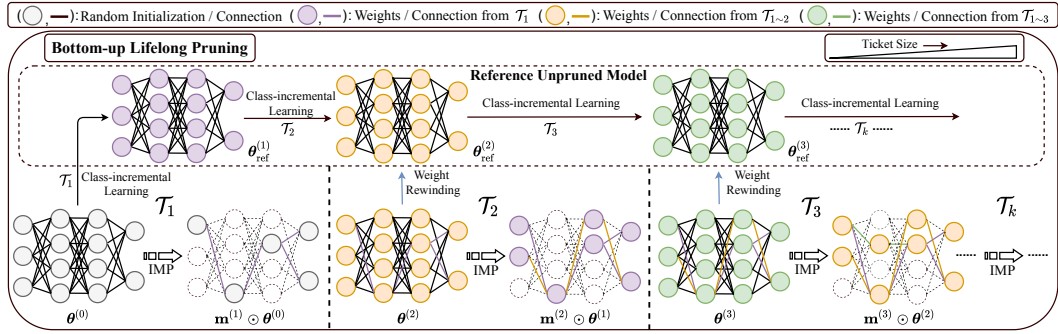

Figure 2: Framework of our proposed bottom-up (BU) lifelong pruning which is based on sparse model consolidation. Tickets founded by BU pruning keep expanding for each newly added task.

**Why we need more than top-down pruning?**    TD pruning is inevitably greedy and suboptimal. Earlier added tasks contribute more to shaping the final mask, due to the nested dependency between intermediate masks. In the later training stage, we often observe the network is already too heavily pruned to learn more tasks. Inspired by the recently proposed model consolidation (Zhang et al., 2020), we propose the BU alternative of lifelong pruning, to dynamically compensate for the excessive pruning by *re-growing* previously reduced networks.

**Full reference model & rewinding point**    For BU lifelong pruning, we maintain a full (unpruned) model $\boldsymbol{\theta}_{\text{ref}}^{(i)}$ as a reference throughout lifelong learning. First, $\boldsymbol{\theta}_{\text{ref}}^{(i)}$ provides a reference performance $\mathcal{R}_{\text{ref}}^{(i)}$ obtained at $\mathcal{T}^{(i)}$. Once the validation accuracy of the current sparse model is no worse than the reference performance, the sparse model is considered to still have sufficient capacity and can be further pruned. Otherwise, capacity expansion is needed. On the other hand, the reference model offers a rewinding point for network parameters, which preserves knowledge of all previous tasks prior to $\mathcal{T}^{(i)}$. It naturally extends the rewinding concept (Frankle et al., 2019) to lifelong learning.

**BU pruning method**    BU lifelong pruning *expands* the previous mask $\boldsymbol{m}^{(i-1)}$ to $\mathbf{m}^{(i)}$. Different from TD pruning, the model size grows along the task sequence, namely, $\|\mathbf{m}^{(i)}\|_0 \geq \|\mathbf{m}^{(i-1)}\|_0$. Thus, BU pruning enforces $\mathcal{A}$ in Eq. (1) to draw non-negative perturbations. As illustrated in Figure 2, for each newly added $\mathcal{T}_i$, we first re-train the previous sparse model $\mathbf{m}^{(i-1)} \odot \boldsymbol{\theta}^{(i-1)}$ under

the current information $(\mathcal{D}^{(i)}, \mathcal{P}^{(i)})$ and calculate the validation accuracy $\mathcal{R}^{(i)}$. If $\mathcal{R}^{(i)}$ is above than the reference performance $\mathcal{R}_{\mathrm{ref}}^{(i)}$, we proceed to keep the sparse mask $\mathbf{m}^{(i)} = \mathbf{m}^{(i-1)}$ intact and use re-trained $\boldsymbol{\theta}^{(i-1)}$ as $\boldsymbol{\theta}^{(i)}$ at $\mathcal{T}_i$. Otherwise, an expansion from $\mathbf{m}^{(i-1)}$ is required to ensure sufficient learning capacity. To do so, we restart from the full reference model $\boldsymbol{\theta}_{\mathrm{ref}}^{(i)}$ and iteratively prune its weights using IMP until the performance gets just below $\mathcal{R}_{\mathrm{ref}}^{(i)}$. Here the previous non-zero weights localized by $\mathbf{m}^{(i-1)}$ are excluded from the pruning scope of IMP but the values of those non-zero weights could be re-trained. As a result, IMP will yield the updated mask $\mathbf{m}^{(i)}$ with a larger size than $\mathbf{m}^{(i-1)}$. We repeat the aforementioned BU pruning method when a new task arrives.

Although never observed in our CIL experiments, a potential corner case of expansion is that the ticket size may hit the size of the full model. We consider this as an artifact of limited model capacity and suggest future work of combining lifelong tickets with (full) model growing (Wang et al., 2017).

**Ticket initialization**   Given the pruning mask found by the BU (or TD) pruning method, we next determine the initialization scheme of a lifelong ticket. We consider *three* specifications of $\boldsymbol{\theta}_0^{(i)}$ in Eq. (1) to initialize the sparse model $\mathbf{m}^{(i)}$ for re-training the found tickets. They include: (I) $\boldsymbol{\theta}_0^{(i)} = \boldsymbol{\theta}^{(0)}$, i.e., the original "from the same random" initialization (Frankle & Carbin, 2019), (II) a random re-initialization $\boldsymbol{\theta}_{\mathrm{reinit}}$ which is independent of $\boldsymbol{\theta}^{(0)}$, and (III) previous-task rewinding, i.e., $\boldsymbol{\theta}_0^{(i)} = \boldsymbol{\theta}^{(i-1)}$. The initialization schemes I-III together with $\mathbf{m}^{(i)}$ yield the following tickets $\mathbf{m}^{(i)}$ at $\mathcal{T}^{(i)}$: (1) BU (or TD) tickets, namely, $\mathbf{m}^{(i)}$ found by BU (or TD) pruning with initialization I; (2) random BU (or TD) tickets, namely, $\mathbf{m}^{(i)}$ with initialization II; (3) task-rewinding BU (or TD) tickets, namely, $\mathbf{m}^{(i)}$ with initialization III. In experiments, we will show that both *BU (or TD) tickets* and their *task-rewinding (TR)* counterparts are winning tickets, which outperform unpruned full CIL models. Compared BU with TD pruning, *TR-BU tickets* surpass the best *TD tickets*.

### 4.3 Lottery Teaching: A Plug-in Regularization

Catastrophic forgetting poses a severe challenge to class-incremental learning, especially for compact models. (Castro et al., 2018; He et al., 2018; Javed & Shafait, 2018; Rebuffi et al., 2017) are early attempts for undertaking the forgetting dilemma by introducing knowledge distillation regularization (Hinton et al., 2015), which employs a handful of stored previous data in addition to new task data. (Zhang et al., 2020) takes advantage of unlabeled data to handle the forgetting quandary.

We adapt their philosophy (Li & Hoiem, 2017; Hinton et al., 2015; Zhang et al., 2020) to presenting lottery teaching, enforcing previous information into the new tickets via a knowledge distillation term on external unlabeled data. Lottery teaching consists of two steps: i) we query more similar unlabeled data "for free" from a public source, by utilizing a small number of prototype samples from previous tasks' training data. In this way, the storage required for previous tasks could be minimal, while the queried surrogate data functions similarly for our purpose; ii) we then enforce the output soft logits of the current subnetwork $\{\mathbf{m}^{(i)} \odot \boldsymbol{\theta}^{(i)}, \boldsymbol{\theta}_{\mathrm{c}}^{(i)}\}$ on each queried unlabeled sample $\mathbf{x}$ to be close to the logits from previously trained subnetwork $\{\mathbf{m}^{(i-1)} \odot \boldsymbol{\theta}^{(i-1)}, \boldsymbol{\theta}_{\mathrm{c}}^{(i-1)}\}$, via knowledge distillation (KD) regularization based on the K-L divergence. For all experiments of our methods hereinafter, we *by default* append the lottery teaching as it is widely beneficial. An ablation study will also follow in Section 5.3.

## 5 Experiments

**Experimental Setup**   We briefly discuss the key facts used in our experiments and refer readers to Appendix A2.3 for more implementation details. We evaluate our proposed lifelong tickets on three datasets: CIFAR-10, CIFAR-100, and Tiny-ImageNet. We adopt ResNet18 (He et al., 2016) as our backbone. We evaluate the model performance by standard testing accuracy (SA) averaged over three independent runs.

CIL baseline: We consider a strong baseline framework derived from (Zhang et al., 2019), a recent state-of-the-art (SOTA) method introduced for imbalanced data training (see more illustrations in Appendix A1.1). We implement (Zhang et al., 2019) for CIL, and compare with two latest CIL

SOTAs: iCaRL (Rebuffi et al., 2017) and IL2M (Belouadah & Popescu, 2019)[1]. Our results demonstrate the adapted CIL method from (Zhang et al., 2019) outperforms the others significantly, ($1.65\%$ SA better than IL2M and $4.88\%$ SA better than iCaRL on CIFAR-10)[2], establishing a new SOTA bar. The proposed *lottery teaching* further improves the performance of the baseline adapted from Zhang et al. (2019), given by $4.4\%$ SA improvements on CIFAR-10. Thus, we use (Zhang et al., 2019), combined with/without *lottery teaching*, to train the original (unpruned) CIL model.

CIL pruning: To the best of our knowledge, we are not aware of any effective CIL pruning baseline comparable to ours. Thus, we focus on the comparison among different variants of our methods. We also compare the proposal with *the ordinary IMP*, showing its incapability in CIL pruning. Furthermore, we demonstrate that given our proposed pruning frameworks, standard pruning methods such as IMP and $\ell_1$ pruning (by imposing $\ell_1$ sparsity regularization) then turn to be successful.

**Results on TD tickets**  We begin by showing that TD pruning is non-trivial in CIL. We find that the ordinary IMP (Han et al., 2015a) fails: It leads to $10.21\%$ SA degradation (from $72.79\%$ to $62.58\%$ for SA) with $4.40\%$ parameters left. By contrast, our proposed lifelong tickets yield substantially better performance which even surpasses the full dense model, with fewer parameters left than the ordinary IMP (Han et al., 2015a).

In what follows, We evaluate TD lifelong pruning using different weight rewindings, namely, i) *TD tickets*; ii) *random TD tickets*; iii) *task-rewinding TD tickets*; iv) *late-rewinding TD tickets*; and v) *Fine-tuning*. The *late-rewinding tickets* is a strong baseline claimed in Mehta (2019).

Figure 3 and Table A4 demonstrate the high competitiveness of our proposed *TD ticket* (blue lines). It matches and most of the time outperforms the full model[3] (black dash lines). Even with only $6.87\%$ model parameters left, the TD ticket still surpasses the dense model by $0.49\%$ SA. The *task-rewinding tickets*, in second place, exceeds the dense model until reaching the extreme sparsity of $4.40\%$. Moreover, we see *late-rewinding TD tickets* dominate over other rewinding/fine-tuning options, echoing the finding in single-task learning (Frankle et al., 2019).

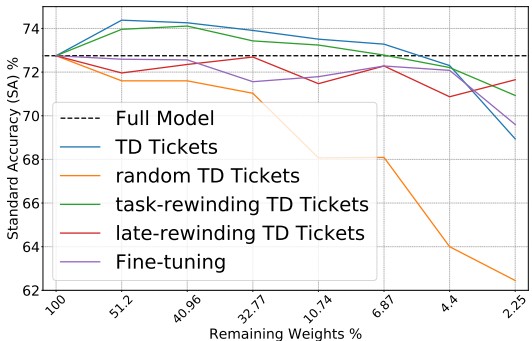

Figure 3: Evaluation performance (standard accuracy) of top-down lifelong tickets on CIFAR-10.

However, TD pruning cannot afford a lot more incremental tasks due to its greedy weight (over-)pruning. Our results show that *TD tickets* pruned from only tasks $\mathcal{T}_1$ and $\mathcal{T}_2$ clearly overfit the first two tasks, even after incrementally learning the remaining three tasks. In this inappropriate pruning schedule (in contrast to $\mathcal{T}_1 \sim \mathcal{T}_5$ scheme), the resultant ticket drops to $59.28\%$ SA which is $13.51\%$ lower than the dense model, as shown in Table A3. More results can be found in the appendix. Therefore, bottom-up lifelong pruning is proposed, as a remedy for relieving laborious tuning of pruning schedules.

**Results on BU lifelong tickets**  The bottom-up lifelong pruning allows the sparse network to regret if they could not deal with the newly added tasks, which compensates for the excessive pruning and reaches a substantially better trade-off between sparsity and generalization ability. Compared to TD pruning, it does not require any heuristic pruning schedules.

In Table 1, we first present the performance of *BU tickets*, *random BU tickets*, and *task-rewinding BU (TR-BU) tickets*, as mentioned in Section 4.2. As we can see, *TR-BU tickets* obtain the supreme performance. A possible explanation is that *task-rewinding* (i.e., $\boldsymbol{\theta}_0^{(i)} = \boldsymbol{\theta}^{(i-1)}$) maintains full information of learned tasks which mitigates the catastrophic forgetting, while other weight rewinding points lack sufficient task information to prevent compact models from forgetting. Next, we observe that *TR-BU tickets* significantly outperform the full dense model by $0.52\%$ SA with only $3.64\%$

---

[1]Both are implemented using official codes. The comparison has been strictly controlled to be fair, including dataset splits, same previously stored data, due diligence in hyperparameter tuning for each, etc.

[2]More comparisons with the latest CIL SOTAs are referred to the Appendix A1.1

[3]Full model, denoting the performance of the dense CIL model (Zhang et al., 2019) with *lottery teaching*.

Table 1: Comparison results across full dense model, *BU Tickets* with different ticket initialization, and $\ell_1$ *BU Tickets* when training incrementally on CIFAR-10. $\mathcal{T}_{1\sim i}$ denotes the learned sequential tasks $\mathcal{T}_1 \sim \mathcal{T}_i$.

| Dataset / Settings | CIFAR-10 (Standard Accuracy / Remaining Weights $\frac{\|\boldsymbol{m}\|_0}{\|\boldsymbol{\theta}\|_0}$) | | | | |
|---|---|---|---|---|---|
| | $\mathcal{T}_1$ (%) / 100% | $\mathcal{T}_{1\sim2}$ (%) / 100% | $\mathcal{T}_{1\sim3}$ (%) / 100% | $\mathcal{T}_{1\sim4}$ (%) / 100% | $\mathcal{T}_{1\sim5}$ (%) / 100% |
| Full Dense Model | 97.75 | 89.10 | 82.83 | 76.99 | 72.79 |
| | $\mathcal{T}_1$ (%) / 2.81% | $\mathcal{T}_{1\sim2}$ (%) / 3.11% | $\mathcal{T}_{1\sim3}$ (%) / 3.40% | $\mathcal{T}_{1\sim4}$ (%) / 3.64% | $\mathcal{T}_{1\sim5}$ (%) / 3.64% |
| *BU tickets* | **98.05** | 86.77 | 75.87 | 70.81 | 68.77 |
| *random BU tickets* | 96.55 | 82.08 | 77.97 | 72.84 | 71.17 |
| *TR-BU tickets* | **98.05** | **88.90** | **81.37** | **74.66** | **73.31** |
| | $\mathcal{T}_1$ (%) / 1.80% | $\mathcal{T}_{1\sim2}$ (%) / 2.93% | $\mathcal{T}_{1\sim3}$ (%) / 2.93% | $\mathcal{T}_{1\sim4}$ (%) / 4.05% | $\mathcal{T}_{1\sim5}$ (%) / 5.16% |
| $\ell_1$ *BU tickets* | 96.80 | 87.05 | 77.58 | 74.53 | 72.88 |

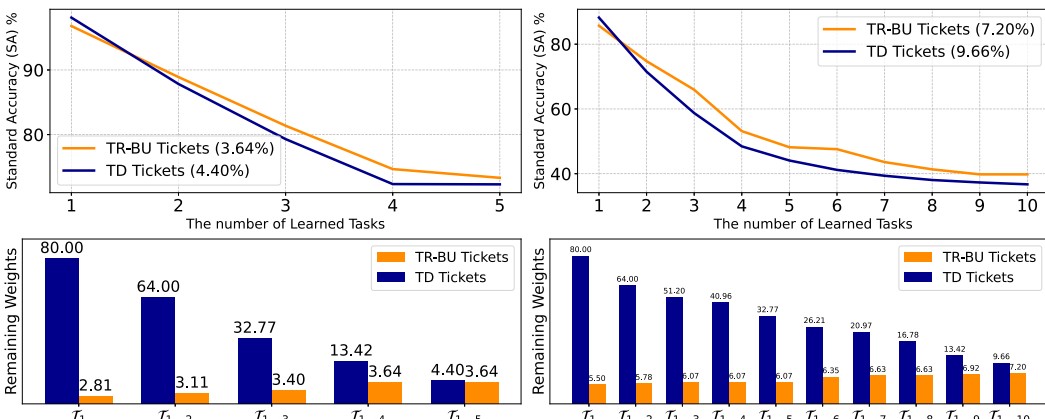

Figure 4: Performance and sparsity comparison between *TR-BU tickets* and *TD tickets* when training models incrementally. Left: CIFAR-10. Right: CIFAR-100. Upper: Comparison of SA. Bottom: Comparison of remaining weights in tickets. Above all, tickets located by TD pruning continue to shrink with the growth of incremental tasks. On the contrary, tickets founded by BU pruning keep expanding for each newly added task.

parameters left and $\ell_1$ *BU tickets* obtain matched performance to the full dense model with $5.16\%$ remaining parameters. It suggests that IMP, $\ell_1$ and even other adequate pruning algorithms can be plugged into our proposed BU pruning framework to identify the lifelong tickets.

In Figure 4, we present the performance comparison between *TR-BU tickets* (the best subnetworks in Table 1) and *TD tickets*. *TR-BU tickets* are identified through bottom-up lifelong pruning, whose sparse masks continue to **subtly grow** along with the incremental tasks, from sparsity $2.81\%$ at the first task to sparsity $3.64\%$ at the last task. As we can see, at any incremental learning stage, *TR-BU tickets* attain a superior performance with significantly fewer parameters. Particularly, after learning all tasks, *TR-BU tickets* surpass *TD tickets* by $1.01\%$ SA with $0.76\%$ fewer weights on CIFAR-10; $3.07\%$ with $2.46\%$ fewer weights on CIFAR-100. Results demonstrate *TR-BU tickets* have a better generalization ability and parameter-efficiency compared with *TD tickets*. In addition, on Tiny-ImageNet in Table A7, *TR-BU tickets* outperform full model with only $12.08\%$ remaining weights. It is worth to mention that both *TR-BU tickets* and *TD tickets* have a superior performance than full dense model. We refer readers to Table A5 and A6 in the appendix for more detailed results.

From the above results, we further observe that *TR-BU tickets* achieve comparable accuracy to full models which have more than $\mathbf{30\times}$ times in network capacity, implying that bottom-up lifelong pruning successfully discovers extremely sparse sub-networks, and yet they are powerful enough to inherit previous knowledge and generalize well on newly added tasks. Furthermore, our proposed lifelong pruning schemes can be directly plugged into other CIL models to identify the lifelong ticket, as shown in Appendix A1.

**Ablation studies** In what follows, we summarize our results on ablation studies and refer readers to the appendix A1.2.1 for more details. In Figure 5, we show the essential role of the curriculum schedule in TD pruning compared to the uniform pruning schedule. We notice that the **curriculum**

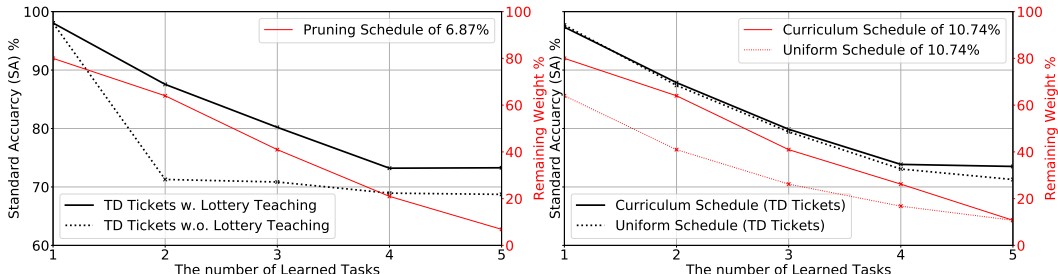

Figure 5: Left: the results of *TD Tickets* with/without lottery teaching. Right: the comparison of *TD tickets* (10.74%) obtained from uniform and curriculum pruning schedule. Experiments are conducted on CIFAR-10.

**pruning scheme** generates stronger *TD tickets* than the uniform pruning in terms of accuracy, which confirms our motivation that pruning heavier in the late stage of lifelong learning with more classes is beneficial. In Table A8, we demonstrate the effectiveness of our proposals against different numbers of incremental tasks. In the Figure 5, we show that **lottery teaching** injects previous knowledge through applying knowledge distillation on external unlabeled data, and greatly alleviates the catastrophic forgetting issue in lifelong pruning (i.e., after learning all tasks, utilizing lottery teaching obtains a $4.34\%$ SA improvement on CIFAR-10). It is worth mentioning that we set a buffer of fixed storage capacity to store $128$ unlabeled images queried from public sources at each training iteration. We find that leveraging newly queried unlabeled data offers a better generalization-ability than storing historical data in past tasks. The latter only reaches $70.60\%$ SA on CIFAR-10, which is $2.19\%$ worse than the use of unlabeled data.

## 6 CONCLUSION

We extend the Lottery Ticket Hypothesis to lifelong learning, in which networks incrementally learn from sequential tasks. We pose top-down and bottom-up lifelong pruning algorithms to identify lifelong tickets. Systematical experiments are conducted to validate that located tickets obtain strong(er) generalization ability across all incremental learned tasks, compared with unpruned models. Our future work aims to explore lifelong tickets with the (full) model growing approach.

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

## A1  MORE EXPERIMENT RESULTS

### A1.1  MORE BASELINE RESULTS

**Comparison with the Latest CIL SOTAs**  We find (Zhang et al., 2019) can be naturally introduced to class-incremental learning, which tackles the intrinsic training bias between a handful of previously stored data and a large amount of newly added data. It adopts random and class-balanced sampling strategies, combined with an auxiliary classifier, to alleviate the negative impact from imbalanced classes. Extensive results, shown in Table A2, demonstrates that adopting (Zhang et al., 2019) as the simple baseline surpasses previous SOTAs iCaRL (Rebuffi et al., 2017) and IL2M (Belouadah & Popescu, 2019) by a significant performance margin (1.65%/0.57% SA better than IL2M and 4.88%/7.60% SA better than iCaRL on CIFAR-10/CIFAR-100, respectively)[4], establishing a new SOTA bar. With the assistance of *lottery teaching*, (Zhang et al., 2019) obtains an extra performance boost, 4.4% SA on CIFAR-10 and 7.34% SA on CIFAR-100.

**It is worth mentioning that a lifelong ticket also exists in other CIL models.**  Take IL2M on CIFAR-10 as an example, bottom-up (BU) ticket achieves accuracy 68.92% with 11.97% parameters vs. the dense unpruned model with an accuracy of 66.74%.

Table A2: Comparison between our dense model and two previous SOTA CIL methods on CIFAR-10 and CIFAR-100. Reported performance it the final accuracy for each task $\mathcal{T}$. Simple baseline donates the dense CIL model (Zhang et al., 2019). Full model represents our proposed framework which combines *lottery teaching* technique with the simple baseline.

| CIFAR-10 Methods | $\mathcal{T}_1$ (%) | $\mathcal{T}_2$ (%) | $\mathcal{T}_3$ (%) | $\mathcal{T}_4$ (%) | $\mathcal{T}_5$ (%) | Average (%) |
|---|---|---|---|---|---|---|
| iCaRL | 76.45 | 79.00 | 75.70 | 50.85 | 35.55 | 63.51 |
| IL2M | 78.20 | 64.05 | 60.40 | 38.95 | 92.10 | 66.74 |
| Simple Baseline | 75.05 | 71.50 | 54.25 | 52.05 | 89.10 | 68.39 |
| Full Model | 79.70 | 79.12 | 68.23 | 63.45 | 73.43 | 72.79 |

| CIFAR-100 Methods | $\mathcal{T}_1$ (%) | $\mathcal{T}_2$ (%) | $\mathcal{T}_3$ (%) | $\mathcal{T}_4$ (%) | $\mathcal{T}_5$ (%) | $\mathcal{T}_6$ (%) | $\mathcal{T}_7$ (%) | $\mathcal{T}_8$ (%) | $\mathcal{T}_9$ (%) | $\mathcal{T}_{10}$ (%) | Average (%) |
|---|---|---|---|---|---|---|---|---|---|---|---|
| iCaRL | 5.90 | 7.50 | 4.50 | 2.80 | 9.00 | 8.00 | 28.20 | 38.50 | 59.60 | 80.20 | 24.42 |
| IL2M | 19.90 | 24.10 | 19.80 | 12.90 | 21.30 | 21.70 | 29.90 | 34.80 | 40.30 | 89.80 | 31.45 |
| Simple Baseline | 21.20 | 32.10 | 23.00 | 22.70 | 21.70 | 31.70 | 39.60 | 33.80 | 40.20 | 54.30 | 32.02 |
| Full Model | 29.04 | 33.94 | 32.54 | 27.94 | 32.74 | 29.64 | 47.94 | 45.34 | 47.24 | 67.24 | 39.36 |

**Pruning Schedule is Important**  As shown in Table A3, an inappropriate pruning schedule across $\mathcal{T}_1 \sim \mathcal{T}_2$, the resultant ticket drops to 59.28% accuracy which is 13.51% lower than the dense model. On the contrary, the adequate scheme across $\mathcal{T}_1 \sim \mathcal{T}_5$ in Table A3, generates a TD winning ticket with a higher test accuracy (+0.49% SA) and extreme fewer parameters (6.87%), compared with the dense CIL model.

Table A3: Evaluation performance of TD tickets (6.87%) pruned from different task ranges.

| Pruning Schedule | TD Tickets (6.87%) on CIFAR-10 | | | | | |
|---|---|---|---|---|---|---|
| | $\mathcal{T}_1$ (%) | $\mathcal{T}_2$ (%) | $\mathcal{T}_3$ (%) | $\mathcal{T}_4$ (%) | $\mathcal{T}_5$ (%) | Average (%) |
| Prune across $\mathcal{T}_1 \sim \mathcal{T}_2$ | 74.05 | 88.80 | 78.25 | 28.40 | 26.90 | 59.28 |
| Prune across $\mathcal{T}_1 \sim \mathcal{T}_5$ | 78.90 | 82.15 | 71.55 | 63.80 | 70.00 | 73.28 |

### A1.2  MORE LIFELONG TICKETS RESULTS

**Top-down Lifelong Tickets**  We also report several performance reference baselines: (a) Full model, denoting the achievable performance of the dense CIL model (Zhang et al., 2019) combined with *lottery teaching*. (b) $CIL_{lower}$ denoting a vanilla CIL model without using lottery teaching

---

[4]To ensure fair compassion, iCaRL and IL2M both are implemented with their official codes. The comparison has been strictly controlled to be fair, including dataset splits, same previously stored data, due diligence in hyperparameter tuning for each, etc.

nor storing/utilizing previous data in any form; (c) $MT_{upper}$ training a dense model using full data from all tasks simultaneously in a multi-task learning scheme. While it is not CIL (and much easier to learn), we consider $MT_{upper}$ as an accuracy "upper bound" for (dense) CIL ; (d) $MT_{LT}$ by directly pruning $MT_{upper}$ to obtain its lottery ticket (Frankle & Carbin, 2019). The detailed evaluation performance of *TD tickets* at different sparsity levels on CIFAR-10 are collected in Table A4.

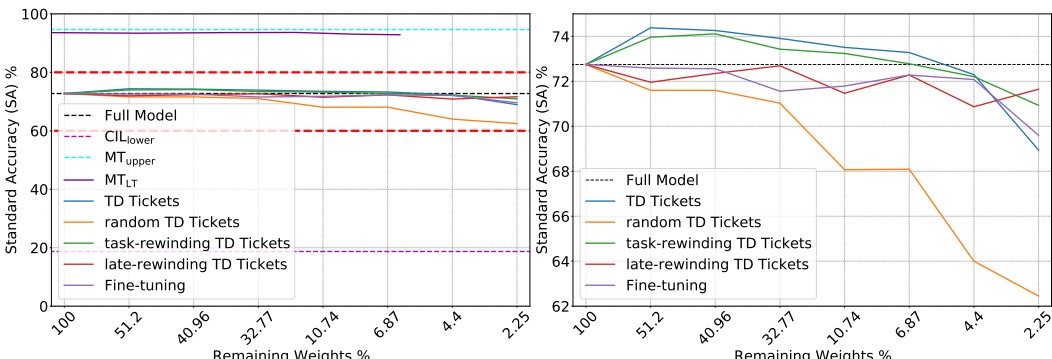

Figure A6: Evaluation performance (standard accuracy) of top-down lifelong tickets. The right figure zooms in the red dash-line box in the left figure.

Table A4: Evaluation performance of *TD tickets* at different sparsity levels on CIFAR-10. Reported performance is the final accuracy for each task $\mathcal{T}$. Differences (+/-) are calculated w.r.t. the full/dense model performance.

| Remaining Weights | TD Tickets on CIFAR-10 | | | | | |
| --- | --- | --- | --- | --- | --- | --- |
| | $\mathcal{T}_1$ (%) | $\mathcal{T}_2$ (%) | $\mathcal{T}_3$ (%) | $\mathcal{T}_4$ (%) | $\mathcal{T}_5$ (%) | Average (%) |
| $MT_{upper}$ (100.00%) | 97.48 | 97.48 | 93.33 | 90.83 | 94.03 | 94.63 |
| $CIL_{lower}$ (100.00%) | 0.00 | 0.00 | 0.00 | 0.00 | 93.70 | 18.74 |
| 100.00% | 79.70 | 79.12 | 68.23 | 63.45 | 73.43 | 72.79 |
| 32.77% | 84.30 | 80.05 | 70.70 | 67.75 | 69.55 | 74.47 + 1.68 |
| 10.74% | 78.90 | 77.75 | 76.30 | 63.55 | 71.05 | 73.51 + 0.72 |
| **6.87%** | 78.90 | 82.15 | 71.55 | 63.80 | 70.00 | 73.28 + 0.49 |
| 4.40% | 82.25 | 78.55 | 65.10 | 65.38 | 70.23 | 72.30 - 0.49 |
| 2.25% | 78.20 | 78.30 | 69.50 | 58.20 | 65.00 | 69.84 - 2.45 |

**Bottom-up Lifelong Tickets**    As shown in Table A5 and Table A6, even compared with the best *TD tickets* in terms of the trade-off between sparsity and accuracy, *TR-BU tickets* consistently remain prominent on both CIFAR-10 (a slightly higher accuracy and $3.23\%$ fewer weights) and CIFAR-100 ($2.37\%$ higher accuracy and $4.88\%$ fewer weights). From the results, we further observe that *TR-BU tickets* achieve comparable accuracy to full models which have more than $\mathbf{30\times}$ times in network capacity, implying that bottom-up lifelong pruning successfully discovers extremely sparse sub-networks, and yet they are powerful enough to inherit previous knowledge and generalize well on newly added tasks.

### A1.2.1    MORE ABLATION RESULTS

**Uniform v.s. Curriculum Lifelong Pruning**    We discuss different pruning schedules of top-down lifelong pruning, which play an essential role in the performance of *TD tickets*. From the right figure in Figure A7, we notice that the curriculum pruning scheme generates stronger *TD tickets* than the uniform pruning in terms of accuracy, which confirms our motivation that pruning heavier in the late stage of lifelong learning with more classes is beneficial.

**The Number of Incremental Tasks**    Here we study the influence of increment times in our lifelong learning settings. Table A8 shows the results of *TR-BU$_{20}$ tickets* incrementally learn from 20 tasks (5

Table A5: Evaluation performance of *TR-BU/TD tickets* on CIFAR-10. $\mathcal{T}_{1\sim i}$, $i \in \{1,2,3,4,5\}$ donates that models have learned from $\mathcal{T}_1, \cdots, \mathcal{T}_i$ incrementally. $\frac{||\boldsymbol{m}||_0}{||\boldsymbol{\theta}||_0}$ represents the current network sparsity.

| Compact Weights | CIFAR-10 | | | | | | | | | |
|---|---|---|---|---|---|---|---|---|---|---|
| | $\mathcal{T}_1$ (%) | $\frac{||\boldsymbol{m}||_0}{||\boldsymbol{\theta}||_0}$ | $\mathcal{T}_{1\sim2}$ (%) | $\frac{||\boldsymbol{m}||_0}{||\boldsymbol{\theta}||_0}$ | $\mathcal{T}_{1\sim3}$ (%) | $\frac{||\boldsymbol{m}||_0}{||\boldsymbol{\theta}||_0}$ | $\mathcal{T}_{1\sim4}$ (%) | $\frac{||\boldsymbol{m}||_0}{||\boldsymbol{\theta}||_0}$ | $\mathcal{T}_{1\sim5}$ (%) | $\frac{||\boldsymbol{m}||_0}{||\boldsymbol{\theta}||_0}$ |
| $\text{MT}_{\text{upper}}$ | - | - | - | - | - | - | - | - | 94.63 | 100.00% |
| $\text{CIL}_{\text{lower}}$ | 97.60 | 100.00% | 49.80 | 100.00% | 32.57 | 100.00% | 23.23 | 100.00% | 18.74 | 100.00% |
| Full Model | 97.75 | 100.00% | 89.10 | 100.00% | 82.83 | 100.00% | 76.99 | 100.00% | 72.79 | 100.00% |
| *TD tickets* (6.87%) | 98.05 | 80.00% | 87.55 | 64.00% | 80.20 | 40.96% | 73.21 | 16.78% | 73.28 | 6.87% |
| *TD tickets* (4.40%) | 98.10 | 80.00% | 87.83 | 64.00% | 79.30 | 32.77% | 72.34 | 13.42% | 72.30 | 4.40% |
| *TR-BU tickets* (3.64%) | 96.80 | 2.81% | 88.90 | 3.11% | 81.37 | 3.40% | 74.66 | 3.64% | **73.31** | **3.64%** |

Table A6: Evaluation performance of *TR-BU/TD tickets* when training incrementally on CIFAR-100.

| Compact Weights | CIFAR-100 | | | | | | | | | |
|---|---|---|---|---|---|---|---|---|---|---|
| | $\mathcal{T}_1$ (%) | $\frac{||\boldsymbol{m}||_0}{||\boldsymbol{\theta}||_0}$ | $\mathcal{T}_{1\sim2}$ (%) | $\frac{||\boldsymbol{m}||_0}{||\boldsymbol{\theta}||_0}$ | $\mathcal{T}_{1\sim3}$ (%) | $\frac{||\boldsymbol{m}||_0}{||\boldsymbol{\theta}||_0}$ | $\mathcal{T}_{1\sim4}$ (%) | $\frac{||\boldsymbol{m}||_0}{||\boldsymbol{\theta}||_0}$ | $\mathcal{T}_{1\sim5}$ (%) | $\frac{||\boldsymbol{m}||_0}{||\boldsymbol{\theta}||_0}$ |
| $\text{MT}_{\text{upper}}$ | - | - | - | - | - | - | - | - | - | - |
| $\text{CIL}_{\text{lower}}$ | 87.40 | 100.00% | 44.65 | 100.00% | 28.67 | 100.00% | 20.08 | 100.00% | 17.44 | 100.00% |
| Full Model | 88.30 | 100.00% | 74.90 | 100.00% | 63.70 | 100.00% | 53.58 | 100.00% | 48.52 | 100.00% |
| TD Tickets (12.08%) | 88.34 | 80.00% | 71.60 | 64.00% | 58.80 | 51.2% | 48.88 | 40.96% | 43.60 | 32.77% |
| TD Tickets (9.66%) | 88.20 | 80.00% | 71.50 | 64.00% | 58.73 | 51.2% | 48.45 | 40.96% | 44.08 | 32.77% |
| BU Tickets (7.20%) | 85.70 | 5.50% | 74.75 | 5.78% | 65.93 | 6.07% | 53.15 | 6.07% | 48.18 | 6.07% |
| Compact Weights | $\mathcal{T}_{1\sim6}$ (%) | $\frac{||\boldsymbol{m}||_0}{||\boldsymbol{\theta}||_0}$ | $\mathcal{T}_{1\sim7}$ (%) | $\frac{||\boldsymbol{m}||_0}{||\boldsymbol{\theta}||_0}$ | $\mathcal{T}_{1\sim8}$ (%) | $\frac{||\boldsymbol{m}||_0}{||\boldsymbol{\theta}||_0}$ | $\mathcal{T}_{1\sim9}$ (%) | $\frac{||\boldsymbol{m}||_0}{||\boldsymbol{\theta}||_0}$ | $\mathcal{T}_{1\sim10}$ (%) | $\frac{||\boldsymbol{m}||_0}{||\boldsymbol{\theta}||_0}$ |
| $\text{MT}_{\text{upper}}$ | - | - | - | - | - | - | - | - | 74.11 | 100.00% |
| $\text{CIL}_{\text{lower}}$ | 14.07 | 100.00% | 12.64 | 100.00% | 10.91 | 100.00% | 9.66 | 100.00% | 8.64 | 100.00% |
| Full Model | 45.82 | 100.00% | 44.07 | 100.00% | 42.35 | 100.00% | 40.93 | 100.00% | 39.36 | 100.00% |
| TD Tickets (12.08%) | 41.50 | 26.21% | 38.19 | 20.97% | 37.28 | 16.78% | 37.63 | 13.42% | 37.42 | 12.08% |
| TD Tickets (9.66%) | 41.18 | 26.21% | 39.37 | 20.97% | 38.06 | 16.78% | 37.30 | 13.42% | 36.72 | 9.66% |
| BU Tickets (7.20%) | 47.58 | 6.35% | 43.59 | 6.63% | 41.35 | 6.63% | 39.80 | 6.92% | **39.79** | **7.20%** |

Table A7: Evaluation performance of *TR-BU/TD tickets* when training incrementally on Tiny-ImageNet.

| Compact Weights | CIFAR-100 | | | | | | | | | |
|---|---|---|---|---|---|---|---|---|---|---|
| | $\mathcal{T}_1$ (%) | $\frac{||\boldsymbol{m}||_0}{||\boldsymbol{\theta}||_0}$ | $\mathcal{T}_{1\sim2}$ (%) | $\frac{||\boldsymbol{m}||_0}{||\boldsymbol{\theta}||_0}$ | $\mathcal{T}_{1\sim3}$ (%) | $\frac{||\boldsymbol{m}||_0}{||\boldsymbol{\theta}||_0}$ | $\mathcal{T}_{1\sim4}$ (%) | $\frac{||\boldsymbol{m}||_0}{||\boldsymbol{\theta}||_0}$ | $\mathcal{T}_{1\sim5}$ (%) | $\frac{||\boldsymbol{m}||_0}{||\boldsymbol{\theta}||_0}$ |
| Full Model | 73.70 | 100.00% | 59.60 | 100.00% | 52.07 | 100.00% | 43.85 | 100.00% | 41.32 | 100.00% |
| BU Tickets (12.08%) | 75.00 | 10.74% | 58.60 | 11.01% | 54.93 | 11.28% | 47.23 | 11.28% | 43.36 | 11.28% |
| Compact Weights | $\mathcal{T}_{1\sim6}$ (%) | $\frac{||\boldsymbol{m}||_0}{||\boldsymbol{\theta}||_0}$ | $\mathcal{T}_{1\sim7}$ (%) | $\frac{||\boldsymbol{m}||_0}{||\boldsymbol{\theta}||_0}$ | $\mathcal{T}_{1\sim8}$ (%) | $\frac{||\boldsymbol{m}||_0}{||\boldsymbol{\theta}||_0}$ | $\mathcal{T}_{1\sim9}$ (%) | $\frac{||\boldsymbol{m}||_0}{||\boldsymbol{\theta}||_0}$ | $\mathcal{T}_{1\sim10}$ (%) | $\frac{||\boldsymbol{m}||_0}{||\boldsymbol{\theta}||_0}$ |
| Full Model | 37.32 | 100.00% | 34.69 | 100.00% | 30.23 | 100.00% | 29.94 | 100.00% | 28.29 | 100.00% |
| BU Tickets (12.08%) | 36.93 | 11.28% | 36.43 | 11.54% | 32.41 | 11.54% | 29.16 | 11.81% | **28.33** | **12.08%** |

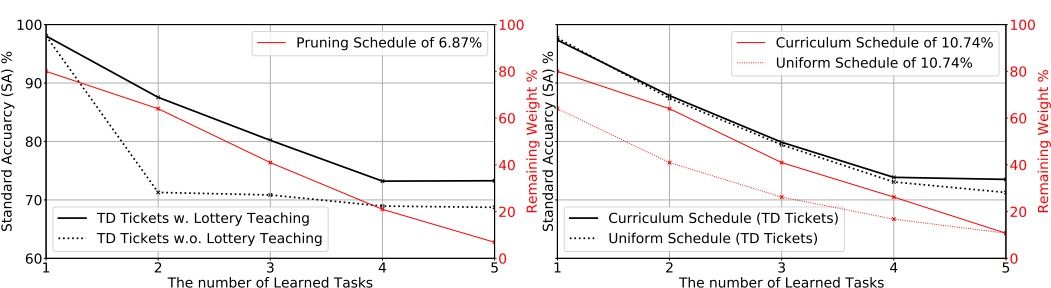

Figure A7: Left: the results of *TD Tickets* with/without lottery teaching. Right: the comparison of *TD tickets* (10.74%) obtained from uniform and curriculum pruning schedule. Experiments are conducted on CIFAR-10.

classes per task); Table A6 presents the results of *TR-BU$_{10}$ tickets* incrementally learn from 10 tasks (10 classes per task). Comparing between two tickets, *TR-BU$_{10}$ tickets* reach $6.55\%$ higher accuracy at the expense of $1.77\%$ more parameters. Possible reasons behind it are that: i) the increasing of incremental learning times aggravates the forgetting issue, which causes *TR-BU$_{20}$ tickets* fall in a

Table A8: Evaluation performance of *TR-BU Tickets* when models incrementally learn **20** tasks on CIFAR-100.

| Compact Weights | CIFAR-100 | | | | | | | | | |
|---|---|---|---|---|---|---|---|---|---|---|
| | $\mathcal{T}_1$ (%) | $\frac{\|\boldsymbol{m}\|_0}{\|\boldsymbol{\theta}\|_0}$ | $\mathcal{T}_{1\sim2}$ (%) | $\frac{\|\boldsymbol{m}\|_0}{\|\boldsymbol{\theta}\|_0}$ | $\mathcal{T}_{1\sim3}$ (%) | $\frac{\|\boldsymbol{m}\|_0}{\|\boldsymbol{\theta}\|_0}$ | $\mathcal{T}_{1\sim4}$ (%) | $\frac{\|\boldsymbol{m}\|_0}{\|\boldsymbol{\theta}\|_0}$ | $\mathcal{T}_{1\sim5}$ (%) | $\frac{\|\boldsymbol{m}\|_0}{\|\boldsymbol{\theta}\|_0}$ |
| Full Model | 89.20 | 100.00% | 73.60 | 100.00% | 67.33 | 100.00% | 59.40 | 100.00% | 52.16 | 100.00% |
| *TR-BU Tickets* (5.43%) | 86.80 | 2.81% | 75.20 | 3.11% | 66.47 | 3.40% | 60.75 | 3.40% | 53.68 | 3.40% |
| Compact Weights | $\mathcal{T}_{1\sim6}$ (%) | $\frac{\|\boldsymbol{m}\|_0}{\|\boldsymbol{\theta}\|_0}$ | $\mathcal{T}_{1\sim7}$ (%) | $\frac{\|\boldsymbol{m}\|_0}{\|\boldsymbol{\theta}\|_0}$ | $\mathcal{T}_{1\sim8}$ (%) | $\frac{\|\boldsymbol{m}\|_0}{\|\boldsymbol{\theta}\|_0}$ | $\mathcal{T}_{1\sim9}$ (%) | $\frac{\|\boldsymbol{m}\|_0}{\|\boldsymbol{\theta}\|_0}$ | $\mathcal{T}_{1\sim10}$ (%) | $\frac{\|\boldsymbol{m}\|_0}{\|\boldsymbol{\theta}\|_0}$ |
| Full Model | 50.10 | 100.00% | 46.80 | 100.00% | 43.35 | 100.00% | 41.71 | 100.00% | 38.62 | 100.00% |
| *TR-BU Tickets* (5.43%) | 50.40 | 3.40% | 47.11 | 3.69% | 44.10 | 3.98% | 43.29 | 4.27% | 38.70 | 4.27% |
| Compact Weights | $\mathcal{T}_{1\sim11}$ (%) | $\frac{\|\boldsymbol{m}\|_0}{\|\boldsymbol{\theta}\|_0}$ | $\mathcal{T}_{1\sim12}$ (%) | $\frac{\|\boldsymbol{m}\|_0}{\|\boldsymbol{\theta}\|_0}$ | $\mathcal{T}_{1\sim13}$ (%) | $\frac{\|\boldsymbol{m}\|_0}{\|\boldsymbol{\theta}\|_0}$ | $\mathcal{T}_{1\sim14}$ (%) | $\frac{\|\boldsymbol{m}\|_0}{\|\boldsymbol{\theta}\|_0}$ | $\mathcal{T}_{1\sim15}$ (%) | $\frac{\|\boldsymbol{m}\|_0}{\|\boldsymbol{\theta}\|_0}$ |
| Full Model | 36.38 | 100.00% | 35.77 | 100.00% | 35.14 | 100.00% | 34.66 | 100.00% | 35.41 | 100.00% |
| *TR-BU Tickets* (5.43%) | 37.35 | 4.27% | 35.88 | 4.27% | 34.62 | 4.27% | 33.87 | 4.27% | 35.48 | 4.56% |
| Compact Weights | $\mathcal{T}_{1\sim16}$ (%) | $\frac{\|\boldsymbol{m}\|_0}{\|\boldsymbol{\theta}\|_0}$ | $\mathcal{T}_{1\sim17}$ (%) | $\frac{\|\boldsymbol{m}\|_0}{\|\boldsymbol{\theta}\|_0}$ | $\mathcal{T}_{1\sim18}$ (%) | $\frac{\|\boldsymbol{m}\|_0}{\|\boldsymbol{\theta}\|_0}$ | $\mathcal{T}_{1\sim19}$ (%) | $\frac{\|\boldsymbol{m}\|_0}{\|\boldsymbol{\theta}\|_0}$ | $\mathcal{T}_{1\sim20}$ (%) | $\frac{\|\boldsymbol{m}\|_0}{\|\boldsymbol{\theta}\|_0}$ |
| Full Model | 34.43 | 100.00% | 33.98 | 100.00% | 34.14 | 100.00% | 32.65 | 100.00% | 33.13 | 100.00% |
| *TR-BU Tickets* (5.43%) | 34.64 | 4.56% | 34.13 | 4.85% | 33.69 | 5.14% | 31.84 | 5.14% | **33.24** | **5.43%** |

worse accuracy decay; ii) at each incremental stage, *TR-BU$_{10}$ tickets* learn more knowledge (10 v.s. 5 classes per task), which requires a large network capacity.

**With v.s. Without Lottery Teaching**  Comparison results between *TD tickets* with lottery teaching and the ones without lottery teaching are collected in this section. As shown in Figure A7 (left figure), the performance of *TD tickets* without lottery teaching (black dash curves), quickly falls into a worse decay along with the times of incremental learning increase. After learning all tasks, utilizing lottery teaching obtains a $4.34\%$ accuracy improvement on CIFAR-10. It suggests that our proposed lottery teaching injects previous knowledge through applying knowledge distillation on external unlabeled data, and greatly alleviates the catastrophic forgetting issue.

## A2  MORE METHODOLOGY AND IMPLEMENTATION DETAILS

### A2.1  MORE LIFELONG PRUNING DETAILS

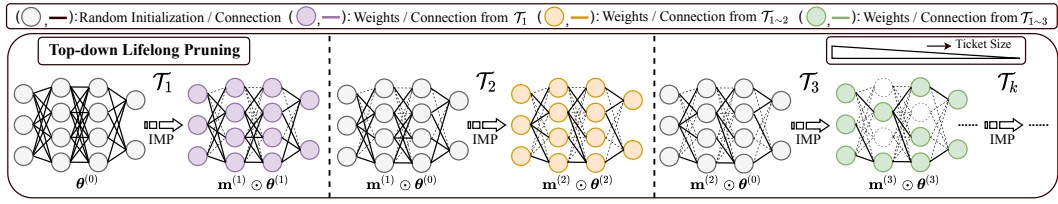

Figure A8: Framework of our proposed top-down lifelong pruning algorithms. The top-down (TD) lifelong pruning performs like iterative magnitude pruning (IMP) by unrolling the sequential tasks. Tickets located by TD pruning continue to shrink with the growth of incremental tasks.

**More Technical Details of Top-down Pruning**  In our implementation, we set $p^{\frac{1}{n^{(i)}}} = 20\%$ as (Frankle & Carbin, 2019; Renda et al., 2020) and adjust $\{n^{(i)}\}$ to control the pruning schedule of IMP over sequential tasks. The aforementioned lifelong pruning method is illustrated in Figure A8, and we call it *top-down* lifelong pruning since the model size is sequentially reduced, namely, $\|\mathbf{m}^{(i)}\|_0 \leq \|\mathbf{m}^{(i-1)}\|_0$.

**Pruning Algorithms**  We summarize the workflow of the top-down pruning and bottom-up pruning in Algorithm 1 and 2, respectively. For pruning hyperparameters, we follow the original LTH's setting (Frankle & Carbin, 2019), i.e. $\Delta p = 20\%$. If we change $\Delta p$ to $40\%$, it will drop $2.04\%$ accuracy at the same sparsity level on CIFAR-10.

### A2.2  MORE CLASS-INCREMENTAL LEARNING DETAILS

**Lottery teaching regularization**  In order to mitigate the catastrophic forgetting effect, we apply knowledge distillation (Hinton et al., 2015) $\mathcal{R}_{\mathrm{KD}}$ to enforce the similarity between previous $\hat{\boldsymbol{y}}$ and

current $\boldsymbol{y}$ soft logits on unlabeled data. We state $\mathcal{R}_{\mathrm{KD}}$ as follows:

$$\mathcal{R}_{\mathrm{KD}}(\boldsymbol{y}, \hat{\boldsymbol{y}}) = -\mathcal{H}(t(\boldsymbol{y}), t(\hat{\boldsymbol{y}}))$$
$$= -\sum_j t(\boldsymbol{y})_j \log t(\hat{\boldsymbol{y}})_j$$

where $t(\boldsymbol{y})_i = \frac{(\boldsymbol{y}_i)^{1/\mathrm{T}}}{\sum_j (\boldsymbol{y}_j)^{1/\mathrm{T}}}$, $\mathrm{T} = 2$ in our case, following the standard setting in (Hinton et al., 2015; Li & Hoiem, 2017).

---

**Algorithm 1:** Top-Down Pruning

**Input:** Full dense model $f(\boldsymbol{\theta}_0, \boldsymbol{\theta}_{\mathrm{c}}^{(0)}; \mathbf{x})$, a desired sparsity $P_m$, samples $\mathbf{x}$ from a storage $\mathcal{S}$ and sequential tasks $\mathcal{T}_{1 \sim n}$, soft logits from previous model on queried unlabeled data, pruning ratio $\Delta p$

**Output:** An updated sparse model $f(\boldsymbol{\theta} \odot \boldsymbol{m}, \boldsymbol{\theta}_{\mathrm{c}}^{(n)}; \mathbf{x})$

1 Set $i = 1$ and mask $\boldsymbol{m} = \mathbf{1} \in \mathbb{R}^{||\boldsymbol{\theta}||_0}$
2 Train $f(\boldsymbol{\theta}_0 \odot \boldsymbol{m}, \boldsymbol{\theta}_{\mathrm{c}}^{(0)}; \mathbf{x})$ with data from $\mathcal{S}$ and $\mathcal{T}_1$.
3 **while** $1 - \frac{||\boldsymbol{m}||_0}{||\boldsymbol{\theta}||_0} \leq P_m$ *and* $i \leq n$ **do**
4     Iterative weight magnitude (IMP) pruning $\Delta p$ and obtaining new mask $\tilde{\boldsymbol{m}}$, where $||\hat{\boldsymbol{m}}||_0 < ||\boldsymbol{m}||_0$
5     Rewind weight to $\boldsymbol{\theta}_0$
6     $\boldsymbol{m} = \hat{\boldsymbol{m}}$
7     Retrain $f(\boldsymbol{\theta}_0 \odot \boldsymbol{m}, \boldsymbol{\theta}_{\mathrm{c}}^{(i)}; \mathbf{x})$ on the current task $\mathcal{T}_i$ and $\mathcal{S}$. Lottery teaching is applied (A knowledge distillation constrain with soft logits)
8     Set $i = i + 1$
9 **end**

---

**Algorithm 2:** Bottom-Up Pruning

**Input:** $f(\boldsymbol{\theta}_0, \boldsymbol{\theta}_{\mathrm{c}}^{(0)}; \mathbf{x})$, $P_m$, $\mathbf{x}$, soft logits and $\Delta p$ defined in Algorithm 1, $f(\boldsymbol{\theta}_i, \boldsymbol{\theta}_{\mathrm{c}}^{(i)}; \mathbf{x})$ has learned $\mathcal{T}_{1 \sim i}$ and has performance $\mathcal{R}_i^*$, $i \in \{1, \cdots, n\}$

**Output:** An updated sparse model $f(\boldsymbol{\theta} \odot \tilde{\boldsymbol{m}}, \boldsymbol{\theta}_{\mathrm{c}}^{(n)}; \mathbf{x})$

1 Set $i = 1$ and mask $\tilde{\boldsymbol{m}} = \mathbf{0} \in \mathbb{R}^{||\boldsymbol{\theta}||_0}$
2 Train $f(\boldsymbol{\theta}_0 \odot \tilde{\boldsymbol{m}}, \boldsymbol{\theta}_{\mathrm{c}}^{(0)}; \mathbf{x})$ with data from $\mathcal{S}$ and $\mathcal{T}_1$. Calculate accuracy $\mathcal{R}_1$.
3 **while** $i \leq n$ *and* $||\tilde{\boldsymbol{m}}||_0 < ||\boldsymbol{\theta}||_0$ **do**
4     **if** $\mathcal{R}_i \geq \mathcal{R}_i^*$ *or* $||\tilde{\boldsymbol{m}}||_0 = ||\boldsymbol{\theta}||_0$ **then**
5        Continue
6     **else**
7        Start from $f(\boldsymbol{\theta}_i, \boldsymbol{\theta}_{\mathrm{c}}^{(i)}; \mathbf{x})$, $\boldsymbol{m} = \mathbf{1}$
8        **repeat**
9           pruning $\Delta p$ of $\boldsymbol{\theta}_i \odot (\boldsymbol{m} - \tilde{\boldsymbol{m}})$, obtain new mask $\boldsymbol{m}^*$, where $||\boldsymbol{m}^*||_0 \geq ||\tilde{\boldsymbol{m}}||_0$ and $\tilde{\boldsymbol{m}} \in \boldsymbol{m}^*$
10           Retrain $f(\boldsymbol{\theta}_{i-1} \odot \boldsymbol{m}^*, \boldsymbol{\theta}_{\mathrm{c}}^{(i)}; \mathbf{x})$ and calculate accuracy $\mathcal{R}_i$
11           $\boldsymbol{m} = \boldsymbol{m}^*$
12        **until** $\mathcal{R}_i \sim \mathcal{R}_i^*$ *and set* $\tilde{\boldsymbol{m}} = \boldsymbol{m}^*$;
13     **end**
14     Set $i = i + 1$
15 **end**

---

**Our Dense Full CIL Model**    We consider a strong baseline framework derived from (Zhang et al., 2019) with our proposed *lottery teaching* as our dense full CIL model. It adopts random and class-balanced sampling strategies, an auxiliary classifier, and the knowledge distillation regularizer $\mathcal{R}_{\mathrm{KD}}$. For incrementally learning task $\mathcal{T}_i$, the training objective is depicted as:

$$\mathcal{L}_{\mathrm{CIL}}(\boldsymbol{\theta}, \boldsymbol{\theta}_c^{(i)}, \boldsymbol{\theta}_a^{(i)}) = \gamma_2 \times \mathcal{L}(\boldsymbol{\theta}, \boldsymbol{\theta}_c^{(i)}) + \mathcal{L}(\boldsymbol{\theta}, \boldsymbol{\theta}_a^{(i)})$$

$$\mathcal{L}(\boldsymbol{\theta}, \boldsymbol{\theta}_c^{(i)}) = \mathbb{E}_{(\mathbf{x}, \mathbf{y}) \in \mathcal{D}_b} \left[ \mathcal{L}_{\mathrm{XE}}(f(\boldsymbol{\theta}, \boldsymbol{\theta}_c^{(i)}, \mathbf{x}), \mathbf{y}) \right]$$
$$+ \gamma_1 \times \mathbb{E}_{\mathbf{x} \in \mathcal{D}_u} \left[ \mathcal{R}_{\mathrm{KD}}(f(\boldsymbol{\theta}, \boldsymbol{\theta}_c^{(i)}, \mathbf{x}), \hat{\boldsymbol{y}}_c) \right],$$

$$\mathcal{L}(\boldsymbol{\theta}, \boldsymbol{\theta}_a^{(i)}) = \mathbb{E}_{(\mathbf{x}, \mathbf{y}) \in \mathcal{D}_r} \left[ \mathcal{L}_{\mathrm{XE}}(f(\boldsymbol{\theta}, \boldsymbol{\theta}_a^{(i)}, \mathbf{x}), \mathbf{y}) \right]$$
$$+ \gamma_1 \times \mathbb{E}_{\mathbf{x} \in \mathcal{D}_u} \left[ \mathcal{R}_{\mathrm{KD}}(f(\boldsymbol{\theta}, \boldsymbol{\theta}_a^{(i)}, \mathbf{x}), \hat{\boldsymbol{y}}_a) \right],$$

where $\mathcal{D}_b$ is the class-balanced sampled dataset, $\mathcal{D}_r$ represents the randomly sampled dataset, and $\mathcal{D}_u$ stands for the queried unlabeled dataset. $\boldsymbol{\theta}_c^{(i)}$ and $\boldsymbol{\theta}_a^{(i)}$ are the main and auxiliary classifiers. $\hat{\boldsymbol{y}}_c$ and $\hat{\boldsymbol{y}}_a$ are soft logits on previous tasks of the main and auxiliary classifiers. We adopt $\gamma_1 = 1, \gamma_2 = 0.5$ in our experiment according to grid search.

### A2.3   MORE OTHER IMPLEMENTATION DETAILS

**Datasets and Task Splittings**   We evaluate our proposed lifelong tickets on CIFAR-10, CIFAR-100, and Tiny-ImageNet datasets, all being standard and state-of-the-art benchmarks for CIL (Krizhevsky & Hinton, 2009). For all three datasets, we randomly split the original training dataset into training and validation with a ratio of $9 : 1$. On CIFAR-10, we divide the 10 classes into splits of 2 classes with a random order ($10/2 = 5$ tasks); On CIFAR-100, we divide the 100 classes into splits of 10 classes with a random order ($100/10 = 10$ tasks); On Tiny-Imagenet, we divide the 200 classes into splits of 20 classes with a random order ($200/20 = 10$ tasks). In this way, when models learn a new incoming task, the dimension of classifiers will increase by 2 for CIFAR-10, 10 for CIFAR-100, and 20 for Tiny-ImageNet. Additionally, 100 images, 10 images, and 5 images per class of learned tasks will be stored for CIFAR-10, CIFAR-100, and Tiny-ImageNet respectively.

**Unlabeled Dataset**   All queried unlabeled data for CIFAR-10/CIFAR-100 are from 80 Million Tiny Image dataset (Torralba et al., 2008), and for Tiny-ImageNet are from ImageNet dataset (Krizhevsky et al., 2012). At each incremental learning stage, $4,500, 450$ and $450$ images per class of learned tasks will be queried, based on the feature similarity with stored prototypes $\{\mathbf{m}^{(i-1)} \odot \boldsymbol{\theta}^{(i-1)}, \boldsymbol{\theta}_c^{(i-1)}\}$ in top-down Pruning and $\{\boldsymbol{\theta}^{(i-1)}, \boldsymbol{\theta}_c^{(i-1)}\}$ in bottom-up Pruning at the $i^{\text{th}}$ CIL stage for CIFAR-10, CIFAR-100, and Tiny-ImageNet respectively. The feature similarity is defined by $\ell_2$ norm distance.

**Training and Evaluation**   Models are trained using Stochastic Gradient Descent (SGD) with $0.9$ momentum and $5 \times 10^{-4}$ weight decay. For 100 epochs training, a multi-step learning rate schedule is conducted, starting from $0.01$, then decayed by 10 times at epochs 60 and 80. During the iterative pruning, we retrain the model for 30 epochs using a fixed learning rate of $10^{-4}$. The batch size for both labeled and unlabeled data is 128. We pick the trained model of the highest validation accuracy and report their performance on the hold-out testing set.

**Other Training Details**   (i) CIFAR-10 and CIFAR-100 can be download at `https://www.cs.toronto.edu/~kriz/cifar.html`. (ii) 80 Million Tiny Image dataset is referred to `http://horatio.cs.nyu.edu/mit/tiny/data/index.html`. (iii) All of our experiments are conducted on NVIDIA GTX 1080-Ti GPUs.

## A3   DISCUSSION

**Challenges of Theoretical Analysis and Future Work**   The theoretical justification of the lottery ticket hypothesis is very limited, except for very shallow networks (Anonymous, 2021). In the meantime, class-incremental learning makes the theoretical analysis more difficult. It is a challenging lifelong learning problem, and the current progress lies in the empirical side rather than the theoretical side. The theoretical analysis is out of scope for this paper and we would like to explore it in the future.

