# OpenReview forum: "Long Live the Lottery: The Existence of Winning Tickets in Lifelong Learning"
_ICLR.cc/2021/Conference — ICLR 2021 Poster_

### Official Review · AnonReviewer3 · 2020-10-20
**Good contributions on the lottery ticket hypothesis to life-long learning with a few open questions**

**Rating:** 8
**Confidence:** 3

**Review:**

[EDIT AFTER DISCUSSIONS] I thank the authors for their answers to my comments. Having read the various threads, I confirm my score and see interesting work in this paper.
[/EDIT]

##########################################################################
Summary: this paper extends the lottery ticket hypothesis to life-long learning. The paper proposes a top-down and bottom-up approach to network pruning and shows that the bottom-up pruning reaches SOTA performance on several datasets while reducing the network size to a few percent of the full model size. The paper also shows higher performance against SOTA for class-incremental learning.

##########################################################################
Reasons for score: To the best of my knowledge, this paper brings novel contributions to the community.  The approach is sound, well-explained, and evaluated rigorously.  The Open Questions section presents open questions but these do not represent a blocker to publication in my opinion.

##########################################################################
Pros: The paper has the following advantages:

- Novelty: to the best of my knowledge, the paper brings a novel contribution to the problem of LTH for life-long learning

- Clarity: the paper is well-structured, clear, and easy to read

- Rigor: the work presented in the paper is rigorous. An ablation study is included and several in-depth analysis are presented. Due diligence has been done on experimental setup. The appendix contains numerous experimentation details (providing code would be even better)

- Impact: the paper brings a significant contribution to the literature by beating or reaching SOTA on lifelong learning.

There are several open questions in the Rebuttal section, which, according to me, should not challenge my score.  However, I am interested in the opinion of the authors and other reviewers on these questions.

##########################################################################
Cons: I do not see major limitations to this paper, apart from the code not being released, which reduces the reproducibility of the paper.  This section contains only minor editorial recommendations.

The first sentence of the abstract should read "The lottery ticket analysis states that..." instead of "demonstrates that..." since a hypothesis cannot "demonstrate" something.  It is a minor detail but since this is the first sentence of the paper, it has a significant impact on the reader's impression of the paper.

Typo page 3 in "can be trained same well in isolation" (this phrase does not make sense)

Typo page 4 in "Why we need beyond top-down pruning" (this phrase does not make sense)

Graphs on Figure 3 (and in the Appendix) are hard to read for small values of remaining weights.  Many scaling the x-axis differently would help.

Typo page 6 in "learning the rest three tasks" (does not make sense)

Page 6, the sentence "Therefore, the bottom-up lifelong pruning debuts, ..." does not make sense

The Related work section is put at the end of the paper, which can make sense (some paper do this regularly).  However, for this particular paper, I would tend to think that moving it up in the paper (close to the beginning) could make sense too.


##########################################################################
Open questions

My understanding when reading page 4 section "Curriculum schedule" is that TD pruning requires the knowledge of the number of tasks.  Is that correct?  How would it extend to an unlimited or unknown number of tasks?

The rewinding point approach seems to require maintaining the full model in parallel to the optimized one.  If that is true, it seems to defeat the purpose of optimizing the model.  Am I missing something?  Also, in the real world, this could have memory implications that could make the approach less practical.

The ticket size seems not to have a theoretical upper bound in this approach.  Is this correct?

Results seem out of noise for most experiments, but it would be nice to have confidence intervals, in particular for the claim that TR-BU outperform the dense model by 0.52% (page 6)

From Figure 4, it seems that the ticket sizes seem to converge for TR-BU and TD as the number of tasks grows.  Is that what is happening?  Any theoretical analysis of this?

#########################################################################

---

> ### Author Response · Authors · 2020-11-20
> **Response to Reviewer #3 [Cons 1 & Open Question 1-5]**
>
> Thank you for the detailed summary. We’re very glad you rate our work as novel, and likewise, we found the set of perceptive questions you raised in your feedback very insightful, pushing us to further improve our paper.
>
> [Cons 1: Inappropriate expression and code release] Thanks for pointing out the inappropriate expressions. And we have fixed them in the modified version. Specifically,
>
> - replace ‘demonstrates’ with ‘states’ in the first sentence of the abstract.
> - modify the expression ‘can be trained same well in isolation’ on Page 3 as ‘can be trained in isolation and reach similar performance as the dense network’.
> - modify the expression ‘Why we need beyond top-down pruning’ on Page 4 as ‘Why we need more than top-down pruning’.
> - scale the x-axis of Figure 3&A6 differently to make it easier to read.
> - modify the expression ‘learning the rest three tasks’ on Page 6 as ‘learning the remaining three tasks’
> - modify the expression ‘Therefore, the bottom-up lifelong pruning debuts, as..’ on Page 6 as ‘Therefore, the bottom-up lifelong pruning is proposed, as...’
> - put the related work section under the introduction.
>
> **As for reproductivity, we will release our code as an additional supplementary material before the end of the rebuttal period.**
>
> [Question 1: TD pruning Explanation] Yes, TD pruning requires knowledge on the number of tasks for designing a good curriculum schedule, and thus might be difficult to extend it to an unlimited or unknown number of tasks. That’s why we proposed Bottom-Up (BU) pruning. Once the current sparse network is too heavily pruned and has no more capacity for new tasks, BU pruning can make the sparse network to re-grow from the current sparsity.
>
> [Question 2: Maintaining Full Model] Thanks for the question. You are right about the memory implications, we agree that the rewinding point approach requires maintaining the full model in parallel. We can not claim the memory efficiency of our methods, but the inference efficiency is still valid since the sparse models can be utilized for predictions.
> Meanwhile, it is fair to note that our main goal is to conduct an extensive and systematic investigation for the lottery ticket hypothesis under CIL settings. We show the existence of lifelong tickets by proposed TD and BU lifelong pruning approaches. Such comprehensive empirical studies are beneficial for a better understanding of deep neural networks.
>
> [Question 3: Theoretical Upper Bound] Thanks for the question. We agree that the ticket size doesn't have a theoretical upper bound in our approach. For example, it is indeed interesting to analyze whether or not BU lifelong tickets would reach the full size as the number of tasks goes to infinity. However, to the best of our knowledge, the theoretical justification of the lottery ticket hypothesis is very limited, except for shallow networks [1], and remains an open question. And the class-incremental learning makes it even more difficult. We hypothesize that BU pruning may be bounded by a  time point after which all classes have been seen in the previous tasks. This could be or close to a full model if the number of classes continuously grows.
>
> [Question 4: Confidence Intervals] Thanks for pointing this out. And we report results over 10 runs: TR-BU ticket (73.31%±0.11% acc. with 3.64%±0.90% parameters) vs. baseline (72.79%±0.08% acc.)  on CIFAR-10.
>
> [Question 5: Ticket Size] Thanks for the question. The ticket size for TR-BU and TD won’t converge to the same value.  In Figure 4, the ticket sizes are maintained to a similar level for a better comparison between TD and BU tickets. Specifically, as for TR-BU tickets, when the number of tasks increases to infinity, the ticket size will gradually grow to the size of the dense model until even the dense model can not handle this CIL task.  By the way, this may be solved with assistance from model growing techniques. As for TD pruning, we use a pre-defined schedule which decides the sparsity of subnetworks.
>
> [1] Why Lottery Ticket Wins? A Theoretical Perspective of Sample Complexity on Sparse Neural Networks

---

> > ### Comment · ~Licette_Griffin1 · 2023-05-30
> > **I love playing lottery games but always fail in winning**
> >
> > I love playing lottery games but always fail in winning. I was researching online one evening when I saw random Testimonies of people appreciating Lord Bubuza for giving them the right lottery winning numbers after casting a lottery spell. I spoke to him on WhatsApp: +1 951 442 2214 for help and he said, I will cast a spell and the lottery winning numbers will be revealed to me by my gods(Oracle). I provided his requirements to cast the spell. After casting the spell, He sent me some numbers and said PLAY WITHOUT FEAR. I bought 2 lottery scratch cards in a local shop and played. I won £1,000,000 pounds on the first one but I didn’t believe so I scratch other ticket and behold I won £10,000,000 pounds again. I immediately retired from my warehouse job when my winning amount hit my account. Lord Bubuza is a great SEER and has changed my financial status. Contact him for help via email: lordbubuzamiraclework @ hotmail . com or WhatsApp: +1 951 442 2214

---

### Official Review · AnonReviewer6 · 2020-11-07
**Long Live the Lottery: The Existence of Winning Tickets in Lifelong Learning**

**Rating:** 7
**Confidence:** 4

**Review:**

##########################################################################

Summary:


The paper provides an interesting extension of the lottery ticket hypothesis in the lifelong learning setup, showing the existence of these tickets for class incremental learning. The paper also explores top-down and bottom-up tickets. The authors performed experiments on CIFAR10,CIFAR100, and Tiny-ImageNet datasets showing the effectiveness of the proposed ticket strategy.


##########################################################################

Reasons for score:


Overall, I am leaning positive. I find the idea of investigating lottery tickets in continual learning is valuable to study and understand. My major concern is about the clarity of the paper and some additional experiments (see cons below). Hopefully, the authors can address my concern in the rebuttal period.


##########################################################################
Pros:


1. The paper focuses on one of the most important machine intelligence tasks, continual learning and more specifically class incremental learning, and studies how to find winning lottery tickets inspired from lottery ticket hypothesis paper (Frankle, Carbin, 2019).


2. The authors proposed top-down and bottom-up tickets as strategies to find these tickets and showed that can perform on bar and sometimes better compared to the full mode.

3. This paper provides experiments on CIFAR10, CIFAR100, min-ImageNet including quantitative results, to show the effectiveness of the proposed initialization.


##########################################################################

Cons:


1. The paper uses episodic memory, a small number of examples from the previous setting. It is not clear whether this observation would generalize to regularization or generative approaches that do not require so. Meaning, lifelong learning methods are desired not to assume access to previous task data (i.e., regularization based approaches like EWS[R1], LWF[R10], Intelligent Synapses[R9], MAS[R3], UCB[R0], generative approaches includes [R6, R13])

2. Related work section can be enriched. There is a lot of work that has been done in continual learning.

a) I attach below some representative references but it will be position this work within regularization, memory-based, and structural continual learning methods.

b) it will be good to also show experimentally how these CIL tickets may generalize in regularization approaches and/or generative approaches mentioned above. Currently, experiments are restricted to iCaRL and IL2M.

3) It seems that iterative CIL pruning requires retraining of the entire sequence. Does not this seem to break the natural setup of revisiting previous task data, again and again even if we prune only once?  I understood that however the goal is to probably show the existence of these CIL tickets but it is still not so clear how to make this more practical and realistic in CIL setting.


##########################################################################

Questions & Clarifications
--------------------------------------
1) continual pruning algorithm A may assign -1,0,1 but some of these weights may not make sense based on m^(i-1). For example, if a mask in some location is already 0, producing -1 is not valid. In the context of this paper, do the authors observe the case of -1, when? will be good to analyze these learning dynamics during the continual learning process.

2) "we find that the schedule of IMP over sequential
tasks, in terms of fn(i)g, is critical to make pruning successful in lifelong learning" could you elaborate on this experimentally? what went wrong in some of your experiments for other choices and why?


3) it could be cleared to update Algorithm 1 and 2 to include the tasks loop.

minor
--------
1) SA is sometimes confusing. In some cases, it is spelled out. sometimes not, will be good to fix it.
2) "Compared BU with TD pruning, TR-BU tickets surpass the best TD tickets."
Comparing? TR-BU abbreviation is not defined before this point but defined later. It will be nice to clarify.

[R1] Kirkpatrick, J., Pascanu, R., Rabinowitz, N., Veness, J., Desjardins, G., Rusu, A. A., ... & Hassabis, D. (2017). Overcoming catastrophic forgetting in neural networks. Proceedings of the national academy of sciences, 114(13), 3521-3526.

[R2]Zenke, Friedemann, Ben Poole, and Surya Ganguli. "Continual learning through synaptic intelligence." Proceedings of machine learning research 70 (2017): 3987.

[R3]Memory Aware Synapses: Learning what (not) to forget. (ECCV’18)
 R Aljundi, F Babiloni, M Elhoseiny, M Rohrbach and T Tuytelaars

[R4]Exploring the Challenges towards Lifelong Fact Learning.  (ACCV’18)
M Elhoseiny, F Babiloni, M Paluri, R Aljundi, M Rohrbach and T Tuytelaars

[R5]David, and Marc'Aurelio Ranzato. "Gradient episodic memory for continual learning." In Advances in neural information processing systems, pp. 6467-6476. 2017.

[R6]Shin, Hanul, Jung Kwon Lee, Jaehong Kim, and Jiwon Kim. "Continual learning with deep generative replay." In Advances in Neural Information Processing Systems, pp. 2990-2999. 2017.

[R7]Rusu, Andrei A., Neil C. Rabinowitz, Guillaume Desjardins, Hubert Soyer, James Kirkpatrick, Koray Kavukcuoglu, Razvan Pascanu, and Raia Hadsell. "Progressive neural networks." arXiv preprint arXiv:1606.04671 (2016).

[R8]Efficient Lifelong Learning with A-GEM  (ICLR’19)
        Arslan Chaudhry, Marc’Aurelio Ranzato, Marcus Rohrbach, Mohamed Elhoseiny

[R9]Uncertainty-guided Continual Learning with Bayesian Neural Networks (ICLR’20)
        Sayna Ebrahimi, Mohamed Elhoseiny, Trevor Darrell, Marcus Rohrbach

[R10] Li, Zhizhong, and Derek Hoiem. "Learning without forgetting." IEEE transactions on pattern analysis and machine intelligence 40.12 (2017): 2935-2947

[R11] Kemker, Ronald, and Christopher Kanan. "Fearnet: Brain-inspired model for incremental learning." arXiv preprint arXiv:1711.10563 (2017).

[R12] Hayes, Tyler L., and Christopher Kanan. "Lifelong machine learning with deep streaming linear discriminant analysis." Proceedings of the IEEE/CVF Conference on Computer Vision and Pattern Recognition Workshops. 2020.

[R13] Liu, Xialei, et al. "Generative Feature Replay For Class-Incremental Learning." Proceedings of the IEEE/CVF Conference on Computer Vision and Pattern Recognition. 2020.

---

> ### Author Response · Authors · 2020-11-20
> **Response to Reviewer #6 [Cons 1-3]**
>
> [Cons 1&2: More Related Works and Experiments] Thanks for the kind suggestions and these representative references. We have enriched the related works with these references in the modified draft and provide comprehensive discussions about regularization, memory-based, and structural continual learning methods.
>
> Besides, in order to verify whether lifelong tickets wound generalize in other CIL methods, we conduct the proposed Bottom-Up (BU) pruning methods in LWF[1]. Table S1 collects the comparison results of a representative regularization based approach (i.e., LWF [1]). We observe that found BU tickets surpass the corresponding full dense model by 1.89% accuracy with only 4.05% remaining weights. It suggests that such lifelong tickets at least can be located in both episodic-memory-based and regularization-based approaches. As for EWC [2], it utilizes the Fisher Information matrix to identify important weights and applies $\ell_2$ weight constraints, which seems to provide an interesting and alternative way for pruning.  We also look into generative methods [3,4]. However, the LTH of generative models in static learning has hardly been explored (e.g., there is one concurrent under-review work of GAN ticket https://openreview.net/forum?id=1AoMhc_9jER). Thus, exploring LTH in continual learning with generative models is much more complicated and can be a separate work on its own. We are willing to continue to investigate the generalization of lifelong tickets with more regularization-based and generative CIL approaches in the future.
>
> Table S1: Comparison results between full dense models and  *BU Tickets* with the **LWF [1] approach** when training incrementally on CIFAR-10. $\mathcal{T}_{1\sim i}$ denotes the learned sequential tasks $\mathcal{T}_1\sim \mathcal{T}_i$.
>
> |Methods|Settings|$\mathcal{T}_1$|$\mathcal{T}_{1\sim2}$|$\mathcal{T}_{1\sim3}$|$\mathcal{T}_{1\sim4}$|$\mathcal{T}_{1\sim5}$|
> |:-:|:-:|:-:|:-:|:-:|:-:|:-:|
> |BU Tickets|Accuracy (%)|97.25|75.98|60.98|48.69|42.37|
> |BU Tickets|Remaining Weights|1.80%|2.93%|2.93%|4.05%|4.05%|
> |Full Dense Model|Accuracy (%)|97.15|77.80|58.60|48.10|40.48|
> |Full Dense Model|Remaining Weights|100%|100%|100%|100%|100%|
>
> [Cons 3: Explanations for Proposed Frameworks] Thanks for the questions. We would like to clarify that our Top-Down (TD) and Button-Up (BU) pruning methods do not require retraining the entire sequence. And in class-incremental continual learning (CIL), when a new task arrives, we no longer have access to the entire dataset of previous tasks. Both iterative lifelong pruning approaches are conducted in the current task, as shown in Figure 2 and A8. Specifically, for TD pruning, we need to define the pruning schedule in advance and conduct the iterative magnitude pruning (IMP) in the current task with pre-defined iteration numbers. For BU pruning, we first train the current sparse network in the new task to verify whether the current sparse network has enough capacity for the new task. If not, we will conduct IMP to the full model on the current task with previous non-zero weights excluded from the pruning scope. Thus our approaches do not require retraining the entire sequence.
>
> [1] Learning without forgetting
>
> [2] Overcoming catastrophic forgetting in neural networks
>
> [3] Continual learning with deep generative replay.
>
> [4] Generative Feature Replay For Class-Incremental Learning.

---

> ### Author Response · Authors · 2020-11-21
> **(Continued) Response to Reviewer #6 [Questions & Clarifications 1-5]**
>
> [Questions & Clarifications 1-5] Thanks a lot for the questions.
>
> - [Invalid mask doesn’t exist] We apologize for the confusion. In our implementation, we won’t assign -1 to the locations which are already 0.  We apply the pruning operation to either 1-valued elements (from 1 to 0 in TD pruning) or 0-valued elements (from 0 to 1 in BU pruning) Specifically, as illustrated in Figure 2 and A8, for TD pruning, the current sparse subnetworks are contained by previous subnetworks since non-zero weights are continually removed; for BU pruning, the current sparse subnetworks contain previous subnetworks since non-zero weights are continually added.
>
> - [curriculum schedule of TD pruning is a key to success] Due to the greedy nature of Top-Down (TD) pruning, the pruning schedule plays an important role. Specifically, when pruning too heavy in the earlier added tasks, the network has no more capacity for learning new ones, which will inevitably exacerbate the catastrophic forgetting effect and cause a significant performance drop. That’s the reason why we proposed Bottom-Up pruning to fix this drawback. In Figure 5 and Section A1.2.1, we compare the TD tickets obtained from uniform and curriculum pruning schedules and notice that the curriculum pruning scheme generates stronger TD tickets. Furthermore, as shown in Table A3, when we assign all pruning budgets during task1 and task2, the final accuracy after incrementally learning all 5 tasks is 59.28%, which means the TD tickets clearly overfit the first two tasks.
>
> - [Paper writing modification for clarification 3-5] Sorry for your confusion about several mislearning expressions in our paper. In our modified draft, we have added a task loop in Algorithm 1&2 and fix the inconsistent expression about SA (standard accuracy) as well as defining TR-BU tickets before using it.

---

### Official Review · AnonReviewer5 · 2020-11-07
**interesting topic, incremental work**

**Rating:** 5
**Confidence:** 3

**Review:**

The research question of this paper is the existence of an extremely sparse network with an initial weight assignment that can be trained online to perform multiple tasks to compete with a dense network, in a lifelong continual learning configuration. Another research question of this paper is how to identify this sparse network and achieve competitive performance. To address these questions, the authors proposed to incrementally introduce new non-zero weights when learning incoming tasks (Figure 2 and Equation 1). The network considered by the authors has a common base for all models and a head for individual tasks.

While the topic that combines lifelong learning and network sparsity is definitely interesting, the development of this paper is incremental,  and there lacks some theoretical justification on why introducing a new task will both keep the network sparse and reuse weights of the previous networks. So the paper is slightly below the acceptance threshold for now.

Pros:

+ Interesting topic.
+ The proposed schema works, as shown in the experiments.

Cons:

- There needs work to satisfactorily define the new lifelong winning ticket framework. For example, in a multi-tasks configuration, why can a new task be learned by adding sparse non-zero weights (+retraining) with the performances of previous tasks preserved? If we simply rely on restarting from scratch when adding sparse new weights doesn't do the work, then what is the contribution? In what situation do we need to restart from scratch and in what situation can new tasks be learned from the current sparse network incrementally?
- Lack of theoretical and experimental justification about how the proposed concept will hold or fail.

 To improve the paper, I would like to see more experiments, and preferably theoretical discussions.

---

> ### Author Response · Authors · 2020-11-18
> **Response to Reviewer #5 [Cons1]**
>
> [Cons 1: Framework explanations] Thanks for the comments!
> Prior to answering the reviewer's questions, we would like to clarify the setting of class-incremental learning (CIL), to which our proposed lottery ticket pruning is applied. In CIL, when a new task arrives, we have no access to the entire dataset of previous tasks (due to limited data storage capacity in continual (life-long) learning ) [1-6], and the class number could increase over time rather than a static classification problem under a fixed number of classes. Based on the above CIL challenges, it is highly non-trivial to tackle the problem of catastrophic forgetting, namely, how to preserve the performance of previous tasks.
>
> Yes, we show that the proposed bottom-up (BU) lottery ticket pruning (namely, properly adding non-zero weights to the existing subnetwork at a new task) helps mitigate the problem of catastrophic forgetting and yields performance improvement over many CIL baselines (Table 1, A5-7 and Section A1.1). Such an improvement is achieved due to two key techniques developed in this paper. First, the existence of lifelong winning tickets (namely, a proper weight mask found by our BU lifelong pruning, together with a proper initialization, e.g., task-rewinding initialization) yields improved model generalization-ability over dense networks (e.g., Figure3 and Table1) in CIL. Note that our result is consistent with previous lottery ticket findings in the static learning setup. Second, the lottery teaching that we introduced in section 3.3 contributes to preserving knowledge from previous tasks; see our ablation study in Figure 5 for justification. As shown in Figure 5, we observe that lottery teaching injects previous knowledge through applying knowledge distillation on external unlabeled data, and greatly alleviates the catastrophic forgetting issue in lifelong pruning (i.e., after learning all tasks, utilizing lottery teaching obtains a 4.34% accuracy improvement on CIFAR-10). Finally, we would like to remark that we did not claim the finding of “the solution” to solve the problem of catastrophic forgetting in CIL.
>
> No, we cannot simply restart from random scratch. That is because, at the current task, we have no access to the entire dataset of previous tasks in CIL. This is one of the main challenges that we encounter and aim to address in this work. As presented in Figure3, random tickets (namely, restart from random initialization) suffer from catastrophic forgetting and incur substantial performance degradation. If the reviewer referred 'scratch' to 'winning lottery ticket', then the catastrophic forgetting issue can largely be alleviated (see our response in the previous paragraph). However, we would like to highlight that it is non-trivial to find the winning ticket. In this paper, we propose BU pruning in CIL to address this issue, as shown in Figure2 and Table1. Specifically, in our BU pruning, if the current sparse subnetwork is capable of learning new tasks incrementally, we keep the sparse subnetwork unchanged; otherwise, we add expand the model capacity by properly adding non-zero weights for a matching performance with respect to the unpruned full CIL model.
>
> [1] Learning without Forgetting
>
> [2] Gradient Episodic Memory for Continual Learning
>
> [3] Insights from the Future for Continual Learning
>
> [4] FearNet: Brain-inspired Model for Incremental Learning
>
> [5] Generative Feature Replay for Class-incremental Learning
>
> [6] IL2M: Class Incremental Learning With Dual Memory

---

> ### Author Response · Authors · 2020-11-18
> **(Continued) Response to Reviewer #5 [Cons2]**
>
> [Cons 2: Lack of theoretical justification and need more experiments]
> Thanks for the suggestions. The theoretical justification of the lottery ticket hypothesis is an open research question. To the best of our knowledge, the theoretical justification is very limited, except for very shallow networks; e.g., the concurrent submission https://openreview.net/pdf?id=8pz6GXZ3YT. In the meantime, class-incremental learning (CIL) makes the theoretical analysis more difficult. It is a challenging lifelong learning problem, and the current progress lies in the empirical side rather than the theoretical side. Based on these, we believe that theoretical justification is out of scope for our work. In the revised paper, we have added a discussion on the challenges of theoretical analysis.
>
> For empirical studies, in our paper, we implement top-down (TD), bottom-up (BU) lifelong pruning methods on three datasets, i.e. CIFAR10, CIFAR100, and Tiny-Imagenet, as presented in Table 1, A5, A6, A7. Extensive experiment results demonstrated the existence of lifelong winning tickets, e.g., achieving 3-8% of the dense model size with higher accuracy, compared to strong class-incremental learning baselines. Moreover, we investigate the impact of different initialization (Table1 and Figure3) and the task-rewinding shows superior performance. We also conduct ablation studies to validate the effectiveness of each proposed component, as presented in Section 4. In the end, we show that a lifelong ticket also exists in other CIL models in Appendix A1.1. Take a representative CIL model, IL2M [6], on CIFAR-10 as an example, BU  ticket achieves accuracy 68.92% with 11.97% parameters vs. the dense unpruned model with accuracy 66.74%.
>
> *If the reviewer could kindly point out what specific experiments that we should further add, we will be glad to try our best to implement them and strengthen our paper during the rebuttal.*
>
> [6] IL2M: Class Incremental Learning With Dual Memory

---

### Public Comment · ~Ghada_Sokar1 · 2020-11-16
**Reference to prior work that uses sparse neural networks for lifelong learning**

Studying the lottery ticket hypothesis in lifelong learning is very interesting and novel work.  The results from the bottom-up pruning approach are very promising. We would like to attract your attention to our prior work SpaceNet: Make Free Space for Continual Learning ( https://arxiv.org/pdf/2007.07617.pdf ) which has a similar idea to the bottom-up lifelong pruning that you proposed. However, we train sparse sub-network from scratch for each task using sparse training instead of pruning. Our paper addresses the class incremental learning paradigm.
It would be nice if you could include a reference to this prior work in the next version of your paper.

---

> ### Author Response · Authors · 2020-11-18
> **Thanks for the reference. We have included it in our modified draft.**
>
> Thank you so much for pointing out the reference. In the paper, instead of lifelong pruning, SpaceNet trains sparse deep neural networks from scratch in an adaptive way that compresses the sparse connections of each task in a compact number of neurons. We have included this paper in the related work and make a further discussion about the difference in our modified draft.

---

### Author Response · Authors · 2020-11-23
**General Response**

We sincerely appreciate all reviewers for rating our work as novel, interesting, and valuable. We truly thank reviewer #3 for the high acknowledgment of our paper's novelty, rigor, and impact. All reviewers’ insightful questions and helpful suggestions are beneficial, pushing us to further strengthen our paper. Before the pointwise responses, we would like to summarize our updates here.

- **[Extra Experiments]** As mentioned by Reviewer #6, we conducted new experiments that apply Bottom-Up pruning to LWF[1] without episodic memory. The superior results demonstrate CIL Tickets can generalize to other lifelong learning methods (e.g., regularization based CIL methods).

- **[Modified Draft]** According to the suggestions from all reviewers, we have updated inappropriate expressions, figures, and algorithms and enriched the related work. The modified draft is updated, and we will keep updating for better and clear readability.

- **[Reproducibility]** we provide both the training and evaluation codes for Top-Down(TD) and Bottom-Up(BU) pruning as additional supplementary material. In addition, the pre-trained BU Winning Tickets and full network can be found at https://www.dropbox.com/sh/4jzu4g83wxn9tgb/AADlIQaAAqTR6MpYj6F1bE23a?dl=0

We hope our pointwise responses below could clarify all reviewers’ confusion. We thank all reviewers’ time again.

[1] learning without forgetting

---

### Comment · ~Albert_Washington1 · 2022-04-17
**HOW I WAS ABLE TO WIN BIG ON MY LOTTERY**

Hi guys, I don’t have much to say right now rather than to say thank you to Dr Amber for rescuing me and my family from poverty. I love playing the lottery but winning big is always the issue for me. I will never forget the day I came in contact with Dr Amber whose lottery spell made me a winner of $360,000,000 million dollars cash prize on the jackpot lottery game I played just by giving me sure winning numbers within 3 days to play the lottery game after he prepared the lottery spell for me. My financial status has changed for good and I’ve started to live my dream life after 14 years of playing the lottery. Thank you Dr Amber for I’m very grateful for all you’ve done for me and my family.  You can TELEGRAM or call or message Dr Amber via +1 (808) 481-5132 or email: amberlottotemple@yahoo.com or visit: amberlottotemple.com for more information.

---

### Comment · ~Mike_Luciano1 · 2022-09-29
**POWERFUL LOTTERY SPELL TO MAKE YOU A MEGA MILLIONAIRE**

Hi everyone... I am Mike Luciano and I’m so addicted to winning the lottery. I’ve just scooped my FOURTH jackpot of $1million – taking my total winnings to $4.6million through the help of one legit spell caster named Dr Amber. My first ever win was $100,000. Last year, I won $500,000 from the Pennsylvania state lottery and I also won $3 million in 2016 bringing the grand total of my winnings to $4.6 million. All my winnings have been made possible with the numbers given to me by Dr Amber. I've been so blessed, winning big three times in my lifetime. His spell casting is unique and safe unlike some fake spell casters that are just after your money without showing any result. Dr Amber is like a God in a human form, he is always there to help and guide me in all my trials in life and with him by my side, I've been living a fulfilled life. Just incase you see my review and needs his help, Email: amberlottotemple@yahoo.com or you call his cell number +18084815132 or reach you out to his website: https://amberlottotemple.com

---

### Comment · ~Licette_Griffin1 · 2023-05-30
**THIS WAS HOW I WON THE LOTTERY**

I love playing lottery games but always fail in winning. I was researching online one evening when I saw random Testimonies of people appreciating Lord Bubuza for giving them the right lottery winning numbers after casting a lottery spell. I spoke to him on WhatsApp: +1 951 442 2214 for help and he said, I will cast a spell and the lottery winning numbers will be revealed to me by my gods(Oracle). I provided his requirements to cast the spell. After casting the spell, He sent me some numbers and said PLAY WITHOUT FEAR. I bought 2 lottery scratch cards in a local shop and played. I won £1,000,000 pounds on the first one but I didn’t believe so I scratch other ticket and behold I won £10,000,000 pounds again. I immediately retired from my warehouse job when my winning amount hit my account. Lord Bubuza is a great SEER and has changed my financial status. Contact him for help via email: lordbubuzamiraclework @ hotmail . com or WhatsApp: +1 951 442 2214

---

### Comment · ~Licette_Griffin1 · 2023-05-30
**ALL THANKS TO LORD BUBUZA**

I love playing lottery games but always fail in winning. I was researching online one evening when I saw random Testimonies of people appreciating Lord Bubuza for giving them the right lottery winning numbers after casting a lottery spell. I spoke to him on WhatsApp: +1 951 442 2214 for help and he said, I will cast a spell and the lottery winning numbers will be revealed to me by my gods(Oracle). I provided his requirements to cast the spell. After casting the spell, He sent me some numbers and said PLAY WITHOUT FEAR. I bought 2 lottery scratch cards in a local shop and played. I won £1,000,000 pounds on the first one but I didn’t believe so I scratch other ticket and behold I won £10,000,000 pounds again. I immediately retired from my warehouse job when my winning amount hit my account. Lord Bubuza is a great SEER and has changed my financial status. Contact him for help via email: lordbubuzamiraclework @ hotmail . com or WhatsApp: +1 951 442 2214

---

### Comment · ~James_wickman1 · 2023-10-06
**BEST WAY TO WIN THE LOTTERY 2023**

MY name is James Wickman from Canada and i won $65 million Lotto Max....
I have been playing lottery games for so many years and I have never won a huge amount of money before but I never give up, I found a website online and I came across so many comments of a great spell caster called DR. BALBOSA who helps so many people to achieve their goals with the lottery winning numbers...I Emailed him and told him all my past experiences and he promise to help me win the lottery and he gave me the lottery winning numbers...
He did a lottery spell and he also pray for me...I was in my vehicle when i heard over the radio that the winning ticket for a $65 million Lotto Max draw was sold in the Timiskaming/Cochrane area so I check my ticket. I was shocked and surprised at that very moment because my long awaited prayers had finally been answered and I thank DR. BALBOSA for helping me out for my financial status to change and that of my generation, contact DR. BALBOSA today from anywhere in the world now and I assure you that your bold step will be your change of level. Email him via: balbosasolutionhome@gmail.com
visit his verified website at https://balbosasolutionhome.com
for immediate response....

---

### Decision · Program_Chairs · 2021-01-07
**Final Decision**

**Decision:**

Accept (Poster)

**Comment:**

This work extends the lottery ticket hypothesis to lifelong learning and, in particular, it tackles the problem of class incremental learning. This is an important and difficult problem, and of great interest to the community. The authors considered top down and bottom-up pruning strategies. The proposed approaches were validated on existing benchmarks (CIFAR10,CIFAR100, and Tiny-ImageNet), reaching state-of-the-art results, and showing that catastrophic forgetting could be alleviated. While some questions remain in terms of practical relevance, they authors showed the existence of winning tickets in the continual setting. There were concerns regarding clarity and requests for additional experiments, but all were convincingly addressed and the clarifications provided by the authors in their rebuttal further strengthened the paper.